# Team-PSRO for Learning Approximate TMECor in Large Team Games via Cooperative Reinforcement Learning

**Stephen McAleer**
Carnegie Mellon University

**Gabriele Farina**
MIT

**Gaoyue Zhou**
Carnegie Mellon University

**Mingzhi Wang**
Peking University

**Yaodong Yang**
Peking University

**Tuomas Sandholm**[*]
Carnegie Mellon University

## Abstract

Recent algorithms have achieved superhuman performance at a number of two-player zero-sum games such as poker and go. However, many real-world situations are multi-player games. Zero-sum two-*team* games, such as bridge and football, involve two teams where each member of the team shares the same reward with every other member of that team, and each team has the negative of the reward of the other team. A popular solution concept in this setting, called TMECor, assumes that teams can jointly correlate their strategies before play, but are not able to communicate during play. This setting is harder than two-player zero-sum games because each player on a team has different information and must use their public actions to signal to other members of the team. Prior works either have game-theoretic guarantees but only work in very small games, or are able to scale to large games but do not have game-theoretic guarantees. In this paper we introduce two algorithms: Team-PSRO, an extension of PSRO from two-player games to team games, and Team-PSRO Mix-and-Match which improves upon Team PSRO by better using population policies. In Team-PSRO, in every iteration both teams learn a joint best response to the opponent's meta-strategy via reinforcement learning. As the reinforcement learning joint best response approaches the optimal best response, Team-PSRO is guaranteed to converge to a TMECor. In experiments on Kuhn poker and Liar's Dice, we show that a tabular version of Team-PSRO converges to TMECor, and a version of Team PSRO using deep cooperative reinforcement learning beats self-play reinforcement learning in the large game of Google Research Football.

## 1 Introduction

Two-player zero-sum games have served as testbeds for artificial intelligence research. Algorithms that have achieved superhuman performance in games such as go (Silver et al., 2017) and poker (Bowling et al., 2015; Brown & Sandholm, 2017) are widely seen as milestones. These algorithms use game-theoretic techniques to find an approximate Nash equilibrium, in other words a strategy that cannot be exploited by humans. Despite these successes in two-player games, real-world scenarios often involve more than two players.

In this paper we study a class of games with multiple players where each player shares reward with the other players in one of two teams, which we call two-team games. A canonical example of a

---

[*]Additional affiliations: Strategy Robot, Inc., Strategic Machine, Inc., Optimized Markets, Inc.

37th Conference on Neural Information Processing Systems (NeurIPS 2023).

two-team game is bridge, a game in which current algorithms fail to compete with expert humans. There are multiple solution concepts for team games, but the one we consider is known as TMECor. In this setting, we allow each team to jointly sample a value from a distribution to correlate their strategies before play. This setting has many nice properties and is the natural solution concept for many team games. Two joint team strategies are in a TMECor if the value that a joint best response for an opponent team gets against each team is the same as the value when both teams are playing their current strategy.

There are two strands of research surrounding two-team games. The first has developed tabular methods that have game-theoretic guarantees. These algorithms work well on very small games but do not scale to large team games. The second has developed deep reinforcement learning algorithms that can scale to large two-team games, but end up being exploitable because they are not based on game-theoretic algorithms.

In this paper, we introduce the first scalable game-theoretic techniques for two-team games. We show that a straightforward extension of PSRO to team games, called Team PSRO, and is guaranteed to converge to an approximate TMECorr. We also introduce a novel algorithm called Team PSRO Mix-and-Match (Team PSRO-MM) that mixes and matches best responses to create a larger population and show that it outperforms Team PSRO. In our experiments, we show that Team PSRO-MM outperforms Team PSRO and that both methods outperform self-play on the large benchmark game of Google Research Football.

Team PSRO is based on *policy space response oracles (PSRO)* (Lanctot et al., 2017). PSRO is already one of the most promising methods for finding approximate Nash equilibrium in large two-player zero-sum games because it is simple to use with existing RL methods, it naturally provides a measure of approximate exploitability, and doesn't require full game-tree traversals. Methods based on PSRO such as AlphaStar (Vinyals et al., 2019b) and Pipeline PSRO (McAleer et al., 2020) have achieved state-of-the-art performance on Starcraft and Barrage Stratego, respectively. Our method, called Team PSRO is the first PSRO-based algorithm for team games and we show that it converges to TMECor when the RL best response is strong enough.

Despite being the first scalable method for computing TMECor, Team PSRO does not efficiently use the population of best responses because each policy must be deployed with the corresponding policy in its joint best response. Building on Team PSRO we fix this issue by allowing policies to be mixed and matched with other policies in the population that are not the corresponding policy in its joint best response. We show that the resulting algorithm, called *Team PSRO Mix-and-Match (Team PSRO-MM)* is able to empirically improve upon Team PSRO. We hypothesize that this is because computing best responses is more costly that evaluating the expected value of policies when the number of policies is low enough.

Our contributions include the following:

1. We show that a straightforward extension of PSRO to team games converges to TMECor.

2. We introduce a novel method based on Team PSRO, called Team PSRO Mix-and-Match, that emperically converges faster by better using the best responses in the population.

## 2   Background

Extensive-form games (EFGs) model games that are played on a game tree, and can capture both sequential and simultaneous moves, as well as private information. In this section, for simplicity we focus on four-player zero-sum games where two players—T1 and T2—play as a team against two opponent players, denoted by O1 and O2. However, these methods generalize to games with more than two players, as shown in our experiments on Google Research Football.

Each node $v$ in the game tree belongs to exactly one player $i \in \{T1, T2, O1, O2\} \cup \{C\}$ whose turn is to move. Player C is a special player, called the *chance player*. It models exogenous stochasticity in the environment, such as drawing a card from a deck or tossing a coin. The edges leaving $v$ represent the actions available at that node. Any node without outgoing edges is called a *leaf* and represents an end state of the game. We denote the set of such nodes by $Z$. Each $z \in Z$ is associated with a tuple of payoffs specifying the payoff $u_i(z)$ of each player $i \in \{T1, T2, O1, O2\}$ at $z$. The product of the probabilities of all actions of C on the path from the root of the game to leaf $z$ is denoted by $p_C(z)$.

Private information is represented via *information set* (infoset). In particular, the set of nodes belonging to $i \in \{\mathsf{T1}, \mathsf{T2}, \mathsf{O1}, \mathsf{O2}\}$ is partitioned into a collection $\mathcal{I}_i$ of non-empty sets: each $I \in \mathcal{I}_i$ groups together nodes that Player $i$ cannot distinguish among, given what they have observed. Necessarily, for any $I \in \mathcal{I}_i$ and $v, w \in I$, nodes $v$ and $w$ must have the same set of available actions. Consequently, we denote the set of actions available at all nodes of $I$ by $A_I$. As it is customary in the related literature, we assume *perfect recall*, that is, no player forgets what he/she knew earlier in the game. Finally, given players $i$ and $j$, two infosets $I_i \in \mathcal{I}_i, I_j \in \mathcal{I}_j$ are *connected*, denoted by $I_i \rightleftharpoons I_j$, if there exist $v \in I_i$ and $w \in I_j$ such that the path from the root to $v$ passes through $w$ or vice versa.

**Sequences.** The set of *sequences* of Player $i$, denoted by $\Sigma_i$, is defined as $\Sigma_i := \{(I, a) : I \in \mathcal{I}_i, a \in A_I\} \cup \{\varnothing\}$, where the special element $\varnothing$ is called the *empty sequence* of Player $i$. The *parent sequence* of a node $v$ of Player $i$, denoted $\sigma(v)$, is the last sequence (information set-action pair) for Player $i$ encountered on the path from the root of the game to that node. Since the game has perfect recall, for each $I \in \mathcal{I}_i$, nodes belonging to $I$ share the same *parent sequence*. So, given $I \in \mathcal{I}_i$, we denote by $\sigma(I) \in \Sigma_i$ the unique parent sequence of nodes in $I$. Additionally, we let $\sigma(I) = \varnothing$ if Player $i$ never acts before infoset $I$.

**Reduced-normal-form plans.** A *reduced-normal-form* plan $\pi_i$ for Player $i$ defines a choice of action for every information set $I \in \mathcal{I}_i$ that is still reachable as a result of the other choices in $\pi$ itself. The set of reduced-normal-form plans of Player $i$ is denoted $\Pi_i$. We denote by $\Pi_i(I)$ the subset of reduced-normal-form plans that prescribe all actions for Player $i$ on the path from the root to information set $I \in \mathcal{I}_i$. Similarly, given $\sigma = (I, a) \in \Sigma_i$, let $\Pi_i(\sigma) \subseteq \Pi_i(I)$ be the set of reduced-normal-form plans belonging to $\Pi_i(I)$ where Player $i$ plays action $a$ at $I$, and let $\Pi_i(\varnothing) := \Pi_i$. Finally, given a leaf $z \in Z$, we denote with $\Pi_i(z) \subseteq \Pi_i$ the set of reduced-normal-form plans where Player $i$ plays so as to reach $z$.

**Sequence-form strategies.** A *sequence-form strategy* is a compact strategy representation for perfect-recall players in EFGs (Romanovskii, 1962; Koller et al., 1996). Given a player $i \in \{\mathsf{T1}, \mathsf{T2}, \mathsf{O1}, \mathsf{O2}\}$ and a normal-form strategy $\mu \in \Delta(\Pi_i)$,[2] the sequence-form strategy induced by $\mu$ is the real vector $\boldsymbol{y}$, indexed over $\sigma \in \Sigma_i$, defined as $y[\sigma] := \sum_{\pi \in \Pi_i(\sigma)} \mu(\pi)$. The set of sequence-form strategies that can be induced as $\mu$ varies over $\Delta(\Pi_i)$ is denoted by $\mathcal{Y}_i$ and is known to be a convex polytope (called the *sequence-form polytope*) defined by a number of constraints equal to $|\mathcal{I}_i|$ (Koller et al., 1996).

**TMECor as a Bilinear Saddle-Point Problem.** A TMECor strategy is a probability distribution $\mu_\mathsf{T}$ over the set of randomized strategy profiles $\mathcal{Y}_\mathsf{T1} \times \mathcal{Y}_\mathsf{T2}$ that guarantees maximum expected utility for the team against the best-responding opponent team $\{\mathsf{O1}, \mathsf{O2}\}$. Since each player has perfect recall, any randomized strategy for a player is equivalent to a distribution over reduced-normal-form pure strategies (Kuhn, 1953). Hence, any distribution over profiles of randomized strategies of the team members can be expressed in an equivalent way as a distribution over *deterministic* strategy profiles $\Pi_\mathsf{T1} \times \Pi_\mathsf{T2}$. Denote by $\mathcal{P}(\Pi_\mathsf{T})$ all possible combinations of pure strategies of a population $\Pi_\mathsf{T}$: $\{(\pi_\mathsf{T1}, \pi_\mathsf{T2}) | \pi_\mathsf{T1}, \pi_\mathsf{T2} \in \Pi_\mathsf{T}\}$. The benefit of this transformation is that $\Pi_\mathsf{T1} \times \Pi_\mathsf{T2}$ is a finite set, unlike $\mathcal{Y}_\mathsf{T1} \times \mathcal{Y}_\mathsf{T2}$. For this reason, TMECor is usually defined in the literature as a distribution over $\Pi_\mathsf{T1} \times \Pi_\mathsf{T2}$ without loss of generality. We will follow the same approach in our characterization.

For each leaf $z$, let $\hat{u}_\mathsf{T}(z) := (u_\mathsf{T1}(z) + u_\mathsf{T2}(z)) p_\mathsf{C}(z)$. The expected utility of the team can be written as the following function of the distributions of play $\mu_\mathsf{T} \in \Delta(\Pi_\mathsf{T1} \times \Pi_\mathsf{T2}), \mu_\mathsf{O} \in \Delta(\Pi_\mathsf{O1} \times \Pi_\mathsf{O2})$:

$$u_\mathsf{T}(\mu_\mathsf{T}, \mu_\mathsf{O}) := \sum_{z \in Z} \hat{u}_\mathsf{T}(z) \left( \sum_{\substack{\pi_\mathsf{T1} \in \Pi_\mathsf{T1}(z) \\ \pi_\mathsf{T2} \in \Pi_\mathsf{T2}(z)}} \mu_\mathsf{T}(\pi_\mathsf{T1}, \pi_\mathsf{T2}) \right) \left( \sum_{\substack{\pi_\mathsf{O1} \in \Pi_\mathsf{O1}(z) \\ \pi_\mathsf{O2} \in \Pi_\mathsf{O2}(z)}} \mu_\mathsf{O}(\pi_\mathsf{O1}, \pi_\mathsf{O2}) \right).$$

For a given opponent coordinated strategy $\mu_\mathsf{O} \in \Delta(\Pi_\mathsf{O1} \times \Pi_\mathsf{O2})$, a *best response* $\mathbb{BR}_\mathsf{T}(\mu_\mathsf{O})$ is defined as a coordinated strategy that gives the highest expected utility against the opponent coordinated strategy:

$$\mathbb{BR}_\mathsf{T}(\mu_\mathsf{O}) = \underset{\mu_\mathsf{T} \in \Delta(\Pi_\mathsf{T1} \times \Pi_\mathsf{T2})}{\arg\max} u_\mathsf{T}(\mu_\mathsf{T}, \mu_\mathsf{O}). \tag{1}$$

By definition, a *team-maxmin equilibrium with coordination device* (TMECor) (Celli & Gatti, 2018; Farina et al., 2018b) is a Nash equilibrium of the game where the team plays according to the

---

[2]$\Delta(X)$ denotes the probability simplex over the finite set $X$.

coordinated strategy $\mu_\mathsf{T} \in \Delta(\Pi_{\mathsf{T1}} \times \Pi_{\mathsf{T2}})$ and the opponent team plays according to the coordinated strategy $\mu_\mathsf{O} \in \Delta(\Pi_{\mathsf{O1}} \times \Pi_{\mathsf{O2}})$. In the zero-sum setting, this amounts to finding a solution of the optimization problem

$$\max_{\mu_\mathsf{T} \in \Delta(\Pi_{\mathsf{T1}} \times \Pi_{\mathsf{T2}})} \min_{\mu_\mathsf{O} \in \Delta(\Pi_{\mathsf{O1}} \times \Pi_{\mathsf{O2}})} u_\mathsf{T}(\mu_\mathsf{T}, \mu_\mathsf{O}). \tag{2}$$

Note that if a pair of joint strategies $(\mu_\mathsf{T}, \mu_\mathsf{O})$ are in a TMECor, then

$$u_\mathsf{T}(\mu_\mathsf{T}, \mu_\mathsf{O}) = u_\mathsf{T}(\mathbb{BR}_\mathsf{T}(\mu_\mathsf{O}), \mu_\mathsf{O}) = u_\mathsf{T}(\mu_\mathsf{T}, \mathbb{BR}_\mathsf{O}(\mu_\mathsf{T})) \tag{3}$$

### 2.1  Approximate TMECor

Define an $\epsilon$-best response ($\epsilon$-BR) $\mathbb{BR}_\mathsf{T}^\epsilon(\mu_\mathsf{O})$ as any coordinated strategy that achieves expected utility against the opponent coordinated strategy within $\epsilon$ of optimal: $u_\mathsf{T}(\mathbb{BR}_\mathsf{T}(\mu_\mathsf{O}), \mu_\mathsf{O}) - \epsilon \leq u_\mathsf{T}(\mathbb{BR}_\mathsf{T}^\epsilon(\mu_\mathsf{O}), \mu_\mathsf{O})$. The *exploitability* $e(\mu_\mathsf{T}, \mu_\mathsf{O})$ of a pair of correlated strategies $(\mu_\mathsf{T}, \mu_\mathsf{O})$ is defined as $e(\mu_\mathsf{T}, \mu_\mathsf{O}) = u_\mathsf{T}(\mathbb{BR}_\mathsf{T}(\mu_\mathsf{O}), \mu_\mathsf{O}) + u_\mathsf{O}(\mu_\mathsf{T}, \mathbb{BR}_\mathsf{O}(\mu_\mathsf{T}))$. A pair of joint strategies $(\mu_\mathsf{T}, \mu_\mathsf{O})$ is in an *$\epsilon$-approximate TMECor* if $e(\mu_\mathsf{T}, \mu_\mathsf{O}) \leq \epsilon$

## 3  Related Work

### 3.1  Double Oracle (DO) and Policy Space Response Oracles (PSRO)

Double Oracle (McMahan et al., 2003) is an algorithm for finding a NE in two-player zero-sum normal-form games. The algorithm works by maintaining a population of strategies for each player. Each iteration a NE is computed for the game restricted to strategies in each player's population. Then, a best response to this NE for each player is computed and added to the population. Although in the worst case DO must expand all pure strategies, in many games DO empirically terminates early and outperforms existing methods. Policy-Space Response Oracles (PSRO) Lanctot et al. (2017) scales DO to large games by using reinforcement learning to approximate a best response. The restricted-game NE is computed on the empirical game matrix generated by having each policy in the population play each opponent policy and tracking average utility in a payoff matrix (Wellman, 2006).

There are a number of methods related to PSRO. NXDO (McAleer et al., 2021; Tang et al., 2023) iteratively adds reinforcement learning policies to a population but solve an extensive-form restricted game and has shown to be more efficient than PSRO in certain games. AlphaStar (Vinyals et al., 2019b) trains a population of policies through a procedure that is somewhat similar to PSRO. AlphaStar also uses some elements of self-play when constructing its population, and outputs a meta-Nash equilibrium of the population at test time. P2SRO (McAleer et al., 2020) parallelizes PSRO with convergence guarantees. Other work has generalized PSRO to more players (Muller et al., 2020; Marris et al., 2021), incorporated diversity objectives to cover more of the strategy space (Liu et al., 2021b; Perez-Nieves et al., 2021; McAleer et al., 2022a; Slumbers et al., 2023; Yao et al., 2023), and meta-learned the meta-distribution (Feng et al., 2021). PSRO has also been applied to core problems in game theory (Zhang et al., 2023) and reinforcement learning (Liang et al., 2023).

### 3.2  Team Games

In this paper we study the TMECor Celli & Gatti (2018); Farina et al. (2018a); Zhang & Sandholm (2021); Zhang et al. (2022) setting where players on the same team are allowed to coordinate before playing. A related solution concept is an equilibrium where team members are not able to coordinate, which is known as a *team-maxmin equilibrium* (TME) strategy (von Stengel & Koller, 1997; Basilico et al., 2017; Zhang & An, 2020a,b; Kalogiannis et al., 2021; Anagnostides et al., 2023). A solution of this type yields the maximum expected utility for the team players against a best-responding opponent. The TMECor solution concept has several advantages over TME. First, the team is guaranteed at least as much expected utility under TMECor than under TME (Celli & Gatti, 2018). Second, the TMECor objective is convex while the TME objective is not convex.

### 3.3  Cooperative Reinforcement Learning

In this section we focus on the regime of *centralized training and decentralized execution (CTDE)* in cooperative reinforcement learning. In the CTDE regime, we assume to have access to all agent's

observations during training time, which is a reasonable assumption for our setting of computational game solving. CTDE methods such as MADDPG Lowe et al. (2017) and COMA Foerster et al. (2018) make use of this centralized training learning a centralized critic that can condition on private information. Although this critic is taken away at test time and only the actor is used in decentralized execution, by conditioning on all relevant information, the centralized critic can reduce much more variance than an individual agent's critic.

Multi-Agent PPO (MAPPO) (Yu et al., 2021; Kuba et al., 2022) trains one PPO agent for the whole team with a centralized critic, among other tricks. Although off-policy CTDE methods such as MADDPG and QMix(Rashid et al., 2018; Schäfer et al., 2023; Mguni et al., 2023) tend to outperform on-policy methods such as COMA (Papoudakis et al., 2020; Wang et al., 2021), in practice MAPPO can be a surprisingly effective and stable baseline, and we use it as our joint best response in Team PSRO.

### 3.4 Deep RL for Team Games

A number of previous works have studied deep reinforcement learning methods in team games. Jaderberg et al. (2019) and Liu et al. (2021a, 2019) train deep RL agents via self play and population based training to achieve high performance on a capture the flag video game and a simulated soccer environment, respectively. Berner et al. (2019a) use self-play reinforcement learning to achieve expert level performance on Dota. None of these methods are grounded in game theoretic techniques, however, and may not be expected to converge to a TMECor. Concurrent to our work, Xu et al. (2023) propose an algorithm that is similar to Team PSRO, but they do not make a connection to TMECor and do not propose the idea of mixing and matching.

## 4 Team PSRO

As described in the previous section, existing methods for learning approximate TMECorrs are tabular, and as a result will not scale to large games. In this paper we propose Team-PSRO. Team-PSRO makes the simple observation that approximate joint best responses can be learned via cooperative reinforcement learning. The resulting algorithm is very similar to PSRO in that every iteration, a best response is computed against the opponent's restricted distribution and the resulting policy is added to a population. However, instead of computing single-agent best responses for a single player, the best responses for Team-PSRO are joint best responses for a single team. These joint best responses are a set of policies for each member of the team and are learned through cooperative reinforcement learning.

### 4.1 Tabular Team Double Oracle

In this section we introduce *team double oracle*, a double oracle method for computing TMECor in zero-sum games with two teams. Team DO is conceptually the same as existing single oracle algorithms (Celli & Gatti, 2018) for team games but extends these algorithms to two team games. Team DO is very similar to the original double oracle algorithm for two-player zero-sum normal form games. Team DO maintains a population of pure joint strategies for each team $\Pi_\mathsf{T}, \Pi_\mathsf{O}$. Each pure strategy in a team's population is a pure joint strategy for each player on that team $\mu_\mathsf{T} = (\pi_{\mathsf{T}1}, \pi_{\mathsf{T}2})$. Similar to DO, we construct a restricted game in the space of pure strategies of the populations for each team. Our restricted game $G(\Pi_\mathsf{T}, \Pi_\mathsf{O})$ is a two-player zero-sum normal-form game that takes as actions the pure strategies $\mu_\mathsf{T} \in \Pi_\mathsf{T}$ for the team and $\mu_\mathsf{O} \in \Pi_\mathsf{O}$ for the opponent team. Every iteration, the NE $(\mu_\mathsf{T}^r, \mu_\mathsf{O}^r)$ of the restricted game $G(\Pi_\mathsf{T}^t, \Pi_\mathsf{O}^t)$ is computed. Next, a best response to this NE in the original game $\mathbb{BR}_\mathsf{O}(\mu_\mathsf{T}^r)$ is computed and added to the population. Team DO terminates when the best response for both players does not improve over the Nash value of the restricted game. Like DO, since there are a finite number of pure strategies, team DO must terminate (potentially in a number of iterations exponential in the size of the game). When it does terminate, the NE of the restricted game is a TMECor of the original game because no best response can improve on the Nash value. Team DO is described in Algorithm 1.

Compared to other methods for finding TMECor, team DO has a number of downsides. First, there are no known convergence rates for double oracle algorithms in general and team DO in particular. In fact, there exist games where Team DO must expand all joint pure strategies of the game before

---

**Algorithm 1** Team Double Oracle (Mix-and-Match)

---

**Result:** TMECor
**Input:** Initial population $\Pi_\mathsf{T}^0, \Pi_\mathsf{O}^0$
**repeat** {for $t = 0, 1, \ldots$}
    **if** Team DO-MM **then**
        $(\mu_\mathsf{T}^r, \mu_\mathsf{O}^r) \leftarrow$ NE in restricted game $G(\mathcal{P}(\Pi_\mathsf{T}^t), \mathcal{P}(\Pi_\mathsf{O}^t))$
    **else**
        $(\mu_\mathsf{T}^r, \mu_\mathsf{O}^r) \leftarrow$ NE in restricted game $G(\Pi_\mathsf{T}^t, \Pi_\mathsf{O}^t)$ {Team DO}
    $\Pi_\mathsf{T}^{t+1} \leftarrow \Pi_\mathsf{T}^t \cup \{\mathbb{BR}_\mathsf{T}(\mu_\mathsf{O}^r)\}$
    $\Pi_\mathsf{O}^{t+1} \leftarrow \Pi_\mathsf{O}^t \cup \{\mathbb{BR}_\mathsf{O}(\mu_\mathsf{T}^r)\}$
**until** $u_\mathsf{T}(\mu_\mathsf{T}^r, \mu_\mathsf{O}^r) = u_\mathsf{T}(\mathbb{BR}_\mathsf{T}(\mu_\mathsf{O}), \mu_\mathsf{O}^r) = u_\mathsf{T}(\mu_\mathsf{T}^r, \mathbb{BR}_\mathsf{O}(\mu_\mathsf{T}))$
**Return:** $(\mu_\mathsf{T}^r, \mu_\mathsf{O}^r)$

---

terminating. Second, in extensive form games, the number of pure joint strategies is exponential in the size of the game. Lastly, computing a best response in team games is computationally intractable in general (Celli & Gatti, 2018).

Despite these issues, team DO is a promising method for the following reasons. First, it naturally outputs a measure of exploitability for free. This is useful to track progress of the algorithm, especially in large team games where computing exploitability is difficult. Second, team DO often terminates well before expanding all joint pure strategies. Indeed, as shown in our experiments below, we find that team DO terminates in a very low number of iterations for all of the games we tested. When team DO terminates in a small number of iterations, it is often much faster than existing methods. Lastly, in the same way that double oracle provides a foundation for PSRO, team DO provides a natural way of scaling up to large games by replacing the best response operator with deep reinforcement learning with Team PSRO. We describe this approach in the next section when we describe our main contribution, Team PSRO.

The main bottleneck for the tabular algorithm consists of the computation of the best response policy of each team. Taking for example the side of team $\mathsf{T}$, that task corresponds to solving the optimization problem

$$\mathbb{BR}_\mathsf{T}(\mu_\mathsf{O}) := \underset{\mu_\mathsf{T} \in \Delta(\Pi_{\mathsf{T}1} \times \Pi_{\mathsf{T}2})}{\arg\max} u_\mathsf{T}(\mu_\mathsf{T}, \mu_\mathsf{O})$$

$$= \underset{\mu_\mathsf{T} \in \Delta(\Pi_{\mathsf{T}1} \times \Pi_{\mathsf{T}2})}{\arg\max} \left\{ \sum_{z \in Z} \hat{u}_\mathsf{T}(z) \left( \sum_{\substack{\pi_{\mathsf{T}1} \in \Pi_{\mathsf{T}1}(z) \\ \pi_{\mathsf{T}2} \in \Pi_{\mathsf{T}2}(z)}} \mu_\mathsf{T}(\pi_{\mathsf{T}1}, \pi_{\mathsf{T}2}) \right) \left( \sum_{\substack{\pi_{\mathsf{O}1} \in \Pi_{\mathsf{O}1}(z) \\ \pi_{\mathsf{O}2} \in \Pi_{\mathsf{O}2}(z)}} \mu_\mathsf{O}(\pi_{\mathsf{O}1}, \pi_{\mathsf{O}2}) \right) \right\}. \quad (4)$$

While the above optimization problem is computationally intractable in the general case (Celli & Gatti, 2018), several computational approaches that can scale up to moderately sized games are known. Celli & Gatti (2018) propose simultaneous row and column generation using an integer-programming-based pricing oracle. Additionally, Farina et al. (2018a); Zhang et al. (2020); Farina et al. (2021) propose approaches based on integer programming, while Zhang & Sandholm (2021); Zhang et al. (2022) propose approaches based on tree decompositions. Finally, Farina & Sandholm (2020) propose a algorithm with worst-case polynomial-time guarantees in a subclass of games called *triangle-free*. As of today, even the best among these approaches can scale only up to games with a few tens of thousands of terminal sequences, and quickly run out of time or memory beyond that scale.

In our tabular experiments, we implemented a variant of the pricing algorithm of Celli & Gatti (2018) to compute best responses for the teams. The algorithm we implemented construct an integer program to capture the domain of feasible $\mu_\mathsf{T}$ in the best-response problem (4). We chose the algorithm by Celli & Gatti (2018) since it only requires a linear number of integer variables in the size of the game tree (this is in contrast, for example, with other integer-programming-based approaches that use a quadratic number of integer variables to achieve a tighter formulation, see e.g. Farina et al. (2021); other approaches such as Zhang & Sandholm (2021); Zhang et al. (2022) use a worst-case exponential number of variables to construct an exact linear programming relaxation of the optimization problem). Pseudocode for the implemented best-response oracle is available in the Appendix.

## 4.2 Tabular Team Double Oracle Mix-and-Match

In this section we introduce *team double oracle mix-and-match (Team DO-MM)*, which builds on Team DO by adding additional joint policies to the population for free. Similarly to Team DO, Team DO-MM maintains a population of pure joint strategies for each team $\Pi_\mathsf{T}, \Pi_\mathsf{O}$. Each pure strategy in a team's population is a pure joint strategy for each player on that team $\mu_\mathsf{T} = (\pi_{\mathsf{T}1}, \pi_{\mathsf{T}2})$. In Team DO-MM, however, we construct a restricted game in an expanded space of pure strategies that considers all possible mixtures of policies from each joint best response. Contrasted with Team DO, Team DO-MM considers pure strategies that mix and match policies from different best responses of the players within each team. Our restricted game $G(\mathcal{P}(\Pi_\mathsf{T}), \mathcal{P}(\Pi_\mathsf{O}))$ is a two-player zero-sum normal-form game that takes as actions the pure strategies $\mu_\mathsf{T} \in \mathcal{P}(\Pi_\mathsf{T})$ for the team and $\mu_\mathsf{O} \in \mathcal{P}(\Pi_\mathsf{O})$ for the opponent team. Similar to Team DO, every iteration, the NE $(\mu_\mathsf{T}^r, \mu_\mathsf{O}^r)$ of the restricted game $G(\mathcal{P}(\Pi_\mathsf{T}^t), \mathcal{P}(\Pi_\mathsf{O}^t))$ is computed. Next, a best response to this NE in the original game $\mathbb{BR}_\mathsf{O}(\mu_\mathsf{T}^r)$ is computed and added to the population. Team DO-MM also terminates when the best response for both players does not improve over the Nash value of the restricted game. Because upon termination no best response can improve on the Nash value, Team DO-MM inherits the convergence guarantee of Team DO to TMECor. Team DO-MM is shown as a one-line change in Algorithm 1.

**Proposition 1.** *Tabular Team DO and Team DO-MM with exact best responses converge to a TMECor.*

*Proof.* Proof is contained in the Appendix. $\square$

## 4.3 Team PSRO

We now describe our main contribution, *team PSRO* and *team PSRO Mix-and-Match*. All existing tabular algorithms cannot scale to a game the size of Google Research Football. And existing algorithms such as self play do not minimize exploitablility, even though they might do well sometimes in practice. Team PSRO scales up Team DO to large games by swapping out an exact joint best response $\mathbb{BR}_\mathsf{T}(\mu_\mathsf{O}^r)$ for an approximate joint best response $\beta_\mathsf{T}$ that is trained through reinforcement learning. Instead of learning a policy for each player individually, as in PSRO, Team PSRO trains a joint best response every iteration. This joint best response is a pair of policies that are trained using cooperative RL. In particular, we use MAPPO to train the joint best response. As in PSRO, once the joint best response for a team is trained, it is added to to that team's population $\Pi_\mathsf{T}^t$ and a two-player zero-sum normal form restricted game $G(\Pi_\mathsf{T}^t, \Pi_\mathsf{O}^t)$ is created. The actions of this normal form game correspond to pure joint strategies in the population and the payoffs correspond to the estimated expected value when one team chooses one joint policy and the other team chooses another joint policy $u_\mathsf{T}(\beta_\mathsf{T}(\mu_\mathsf{O}), \mu_\mathsf{O}^r)$. Payoff values in the restricted game are estimated by sampling rollouts from both teams' joint policies. Team PSRO is described in Algorithm 2.

Note that when the joint best responses achieve expected value within $\epsilon$ of the optimal best response, then upon convergence Team PSRO will output an approximate TMECor. Even when the cooperative reinforcement learning is not within $\epsilon$ of the optimal best response, we find that in practice the reward that joint best responses achieve against the restricted NE tends to go down over time. We call this evaluation metric of the sum of the reward that both joint best responses achieve against the restricted NE *approximate exploitability* because the this is a lower bound on the exact exploitability. The better the cooperative RL algorithm is in team PSRO, the more accurately the approximate exploitability will match true exploitability. As mentioned previously, a key benefit of Team PSRO is that it outputs approximate exploitability as a byproduct of the algorithm. If the RL algorithm is very strong and Team PSRO converges, then one can be reasonably sure that the strategy produced is strong.

The MAPPO algorithm we use contains a centralized critic value function that has access to the history of the game. Since we train the joint best response centrally but execute decentrally, MAPPO allows the critic to see the partner's cards at training time to reduce variance in the update. Another MAPPO trick we use is to share the parameters of the neural network between partners in a joint best response. This has been found to help stabilize training. Although we use MAPPO as our joint RL best response, in principle any cooperative RL algorithm could be used to learn an approximate joint best response.

**Algorithm 2** Team PSRO (Mix-and-Match)

---

**Result:** Approximate TMECorr
**Input:** Initial population $\Pi_\mathsf{T}^0, \Pi_\mathsf{O}^0$
**repeat** {for $t = 0, 1, \ldots$}
  **if** Team-PSRO-MM **then**
    $(\mu_\mathsf{T}^r, \mu_\mathsf{O}^r) \leftarrow$ NE in restricted game $G(\mathcal{P}(\Pi_\mathsf{T}^t), \mathcal{P}(\Pi_\mathsf{O}^t))$
  **else**
    $(\mu_\mathsf{T}^r, \mu_\mathsf{O}^r) \leftarrow$ NE in restricted game $G(\Pi_\mathsf{T}^t, \Pi_\mathsf{O}^t)$
  **for** $m$ iterations **do**
    Update joint best response $\beta_\mathsf{T}$ toward $\mathbb{BR}_\mathsf{T}(\mu_\mathsf{O}^r)$ via cooperative RL
    Update joint best response $\beta_\mathsf{O}$ toward $\mathbb{BR}_\mathsf{O}(\mu_\mathsf{T}^r)$ via cooperative RL
  $\Pi_\mathsf{T}^{t+1} \leftarrow \Pi_\mathsf{T}^t \cup \{\beta_\mathsf{T}\}$
  $\Pi_\mathsf{O}^{t+1} \leftarrow \Pi_\mathsf{O}^t \cup \{\beta_\mathsf{O}\}$
**until** $u_\mathsf{T}(\beta_\mathsf{T}(\mu_\mathsf{O}), \mu_\mathsf{O}^r) + u_\mathsf{O}(\mu_\mathsf{T}^r, \beta_\mathsf{O}(\mu_\mathsf{T})) \leq u_\mathsf{T}(\mu_\mathsf{T}^r, \mu_\mathsf{O}^r) + u_\mathsf{O}(\mu_\mathsf{T}^r, \mu_\mathsf{O}^r) + \epsilon$
**Return:** $(\mu_\mathsf{T}^r, \mu_\mathsf{O}^r)$

---

### 4.4 Team PSRO Mix-and-Match

Similarly to how Team DO-MM builds on Team DO by adding mixed-and-matched pure strategies, Team PSRO Mix-and-Match (Team PSRO-MM) builds on Team PSRO by adding mixed-and-matched deep RL policies. As in Team PSRO, once the joint best response for a team is trained, it is added to to that team's population $\Pi_\mathsf{T}^t$. Similarly to Team DO-MM, an expanded two-player zero-sum normal form restricted game $G(\mathcal{P}(\Pi_\mathsf{T}^t), \mathcal{P}(\Pi_\mathsf{O}^t))$ is created. The actions of this normal form game correspond to mixed-and-matched joint strategies in the population and the payoffs correspond to the estimated expected value when one team chooses one joint policy and the other team chooses another joint policy $u_\mathsf{T}(\beta_\mathsf{T}(\mu_\mathsf{O}), \mu_\mathsf{O}^r)$. Payoff values in the restricted game are estimated by sampling rollouts from both teams' joint policies. Team PSRO-MM is described in Algorithm 2.

## 5 Experiments

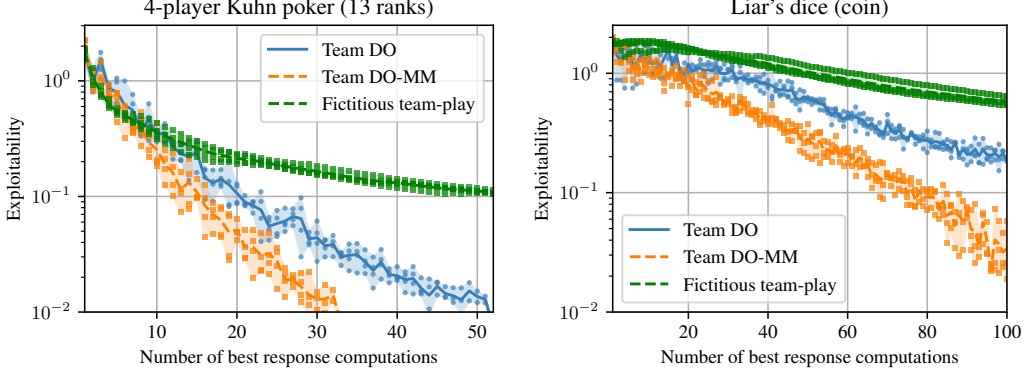

Figure 1: Experimental results in tabular games

### 5.1 Team DO Experiments

We run Team DO and Team DO-MM on two small games, Kuhn poker and Liar's dice, and present the results in Figure 1. Although we do not know of any existing regret bounds for double oracle techniques, these results demonstrate that in practice double oracle methods work well and can find approximate TMECor in a reasonable number of iterations. Notably, Team DO-MM is able to achieve faster convergence compared to Team DO in the number of iterations. However, we found that since calculating best responses in this game does not take much time, computing the extra payoff values caused Team DO-MM to converge much slower in wall clock time compared to Team DO.

Additionally, both Team-DO methods outperform Fictitious Team Play (Farina et al., 2018b), which adds team best responses to the opponent average strategy every iteration. In large games, as shown below, finding a best response requires much more time than evaluating the expected value of two team policies, so Team DO-MM converges faster in wall clock time as well. We include a detailed analysis of wall clock time on these tabular experiments in the appendix.

## 5.2 Team PSRO Experiments

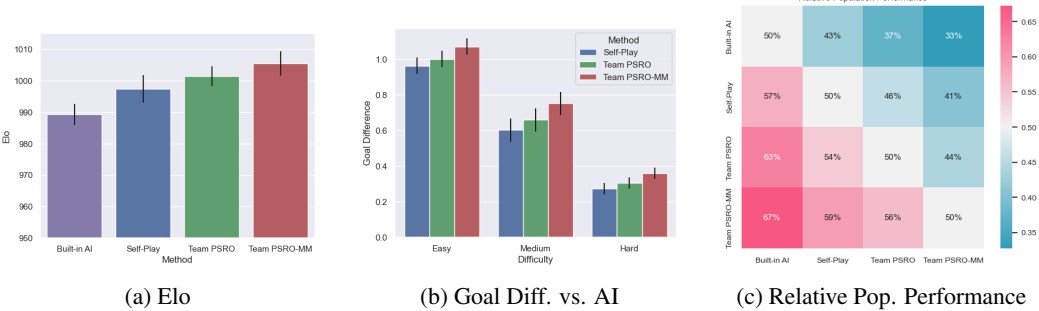

(a) Elo                    (b) Goal Diff. vs. AI          (c) Relative Pop. Performance

Figure 2: Team PSRO Results on Google Research Football. Both Team PSRO and Team PSRO-MM outperform self play, with Team PSRO-MM performing the best.

We demonstrate that Team PSRO can scale to large games by training it on the benchmark game of Google Research Football (GRF) (Kurach et al., 2020). Because there do not exist any other game-theoretic techniques for approaching TMECor in large games, we only compare to self play. Although we do not know of any theoretical guarantees that self play will converge to TMECor, methods based on self play (Berner et al., 2019b; Liu et al., 2019; Jaderberg et al., 2019) have achieved impressive performance on large team games. As a result, we believe that self-play reinforcement learning is a strong benchmark to beat. In these experiments we use the four-player version of GRF and modify it so that each player only observes the information of his own team, the ball, and three closest opponents' information (which means that there are always two opponents that are not observable for a player). We also include results on the perfect-information version of the game in the Appendix and find that the same pattern holds where Team PSRO-MM is better than Team PSRO which is better than self-play. We also include further training details in the Appendix.

We use the same reinforcement learning algorithm for both Team PSRO and self play, namely MAPPO with a centralized critic and shared parameters for a team. We allow the centralized critic to see all players' information. Self play reinforcement learning continually trains one joint policy for one team against the other joint policy for the other team and does not make use of a population like Team PSRO. In Figure 2, we see that Team PSRO and Team PSRO-MM are both able to outperform self-play across all three measures of Elo, average goal difference against the built-in AI, and relative population performance (Balduzzi et al., 2019). Furthermore, Team PSRO-MM outperforms Team PSRO with a small additional cost of extra rollouts to evaluate the matrix payoffs. Since this is such a large game, the time needed to compute the BRs is much more than the time needed to evaluate two team policies, so the extra time is negligible.

## 6 Discussion

### 6.1 Limitations

As with any double oracle algorithm, the main limitation of our algorithm is that it suffers from a lack of practical guarantees. It is possible that all pure strategies of the game must be expanded before termination, and indeed we find that in Kuhn poker, even using an exact best response plateaus indefinitely. With that being said, sometimes exploitability can be too strict of a measure if the exact best response is hard to find. In the case of team games, where finding an exact best response is NP-Hard, perhaps being approximately unexploitable against a reinforcement learning best response is a reasonable goal, and one that we achieve with Team-PSRO.

## 6.2 Future Work

There are many future directions to build on this work. First, we are interested in continuing to scale up to large games, with the goal of becoming superhuman at bridge. A second direction of future work involves studying the problem of cooperative reinforcement learning directly, where improvements should directly transfer to our setting. A third research direction is transferring over new PSRO algorithms such as Anytime PSRO (McAleer et al., 2022b), and NXDO (McAleer et al., 2021). Related efforts in solving team games using regret minimization and subgame solving are possibly complimentary to our approach.

# 7    Acknowledgements

This material is based on work supported by the Vannevar Bush Faculty Fellowship ONR N00014-23-1-2876, National Science Foundation grants RI-2312342 and RI-1901403, ARO award W911NF2210266, and NIH award A240108S001.

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

## A Proof of Proposition 1

*Proof.* Note that by Equation 3 and the stopping condition in Algorithm 1, when Team DO(-MM) terminates, the meta-NE $(\mu_T^r, \mu_O^r)$ is a TMECor. What is left to show is that Team DO(-MM) eventually terminates. Team DO(-MM) eventually terminates because in every iteration it either adds a new strategy for at least one player or it terminates. To see this, consider the case where no new strategies are added for either player in an iteration. Then the best response value for each player is no greater than the value of their meta-NE against the opponent, so Equation 3 is satisfied and Team DO(-MM) terminates in a TMECor. □

## B Tabular Best-Response Oracle

In this section we give the formulation of the team-best-response oracle we implemented in the tabular case.

As discussed in the body (Section 4.1), the algorithm is based on integer programming and implements the algorithm introduced by Celli & Gatti (2018). The integer programming formulation uses the

fact that by convexity, the optimal solution $\mu^* \in \Delta(\Pi_{\mathsf{T}1} \times \Pi_{\mathsf{T}2})$ in (4) puts all mass on exactly a pair $(\pi_{\mathsf{T}1}^*, \pi_{\mathsf{T}2}^*) \in \Pi_{\mathsf{T}1} \times \Pi_{\mathsf{T}2}$. With that, for any $z \in Z$ we can write

$$\sum_{\substack{\pi_{\mathsf{T}1} \in \Pi_{\mathsf{T}1}(z) \\ \pi_{\mathsf{T}2} \in \Pi_{\mathsf{T}2}(z)}} \mu_{\mathsf{T}}^*(\pi_{\mathsf{T}1}, \pi_{\mathsf{T}2}) = \mathbf{1}[\pi_{\mathsf{T}1}^* \in \Pi_{\mathsf{T}1}(z)] \cdot \mathbf{1}[\pi_{\mathsf{T}2}^* \in \Pi_{\mathsf{T}2}(z)] = \pi_{\mathsf{T}1}^*[\sigma_{\mathsf{T}1}(z)] \cdot \pi_{\mathsf{T}2}^*[\sigma_{\mathsf{T}2}(z)].$$

Hence, we can rewrite (4) as

$$\mathbb{BR}_{\mathsf{T}}(\mu_{\mathsf{O}}) = \operatorname*{arg\,max}_{(\pi_{\mathsf{T}1}^*, \pi_{\mathsf{T}2}^*) \in \Pi_{\mathsf{T}1} \times \Pi_{\mathsf{T}2}} \left\{ \sum_{z \in Z} \hat{u}_{\mathsf{T}}(z) \pi_{\mathsf{T}1}^*[\sigma_{\mathsf{T}1}(z)] \pi_{\mathsf{T}2}^*[\sigma_{\mathsf{T}2}(z)] \left( \sum_{\substack{\pi_{\mathsf{O}1} \in \Pi_{\mathsf{O}1}(z) \\ \pi_{\mathsf{O}2} \in \Pi_{\mathsf{O}2}(z)}} \mu_{\mathsf{O}}(\pi_{\mathsf{O}1}, \pi_{\mathsf{O}2}) \right) \right\}.$$

We can represent the set of plans $(\pi_{\mathsf{T}1}^*, \pi_{\mathsf{T}2}^*) \in \Pi_{\mathsf{T}1} \times \Pi_{\mathsf{T}2}$ using binary variables and linear constraints, by using the well-known fact that $\Pi_{\mathsf{T}1} = \mathcal{Y}_{\mathsf{T}1} \cap \{0,1\}^{\Sigma_{\mathsf{T}1}}$, $\Pi_{\mathsf{T}2} = \mathcal{Y}_{\mathsf{T}2} \cap \{0,1\}^{\Sigma_{\mathsf{T}2}}$. Finally, for each $z \in Z$ we model the product $\pi_{\mathsf{T}1}^*[\sigma_{\mathsf{T}1}(z)] \pi_{\mathsf{T}2}^*[\sigma_{\mathsf{T}2}(z)]$ in the objective function by introducing additional binary variables $\xi_z$ subject to the constraints (i) $\xi_z \le \pi_{\mathsf{T}1}^*[\sigma_{\mathsf{T}1}(z)]$, (ii) $\xi_z \le \pi_{\mathsf{T}2}^*[\sigma_{\mathsf{T}2}(z)]$, and (iii) $\xi_z \ge \pi_{\mathsf{T}1}^*[\sigma_{\mathsf{T}1}(z)] + \pi_{\mathsf{T}2}^*[\sigma_{\mathsf{T}2}(z)] - 1$. Putting everything together, we arrive to the best-response integer program

$$\begin{cases} \arg\max \quad \displaystyle\sum_{z \in Z} \hat{u}_{\mathsf{T}}(z) \xi_z \left( \sum_{\substack{\pi_{\mathsf{O}1} \in \Pi_{\mathsf{O}1}(z) \\ \pi_{\mathsf{O}2} \in \Pi_{\mathsf{O}2}(z)}} \mu_{\mathsf{O}}(\pi_{\mathsf{O}1}, \pi_{\mathsf{O}2}) \right), \quad \text{subject to:} \\[2em] \text{①}\ \xi_z \le \pi_{\mathsf{T}1}^*[\sigma_{\mathsf{T}1}(z)] \qquad \forall z \in Z \\[0.5em] \text{②}\ \xi_z \le \pi_{\mathsf{T}2}^*[\sigma_{\mathsf{T}2}(z)] \qquad \forall z \in Z \\[0.5em] \text{③}\ \xi_z \ge \pi_{\mathsf{T}1}^*[\sigma_{\mathsf{T}1}(z)] + \pi_{\mathsf{T}2}^*[\sigma_{\mathsf{T}2}(z)] - 1 \qquad \forall z \in Z \\[0.5em] \text{④}\ \pi_{\mathsf{T}1} \in \mathcal{Y}_{\mathsf{T}1} \\[0.5em] \text{⑤}\ \pi_{\mathsf{T}2} \in \mathcal{Y}_{\mathsf{T}2} \\[0.5em] \text{⑥}\ \xi_z \in \{0,1\} \qquad \forall z \in Z \\[0.5em] \text{⑦}\ \pi_{\mathsf{T}1} \in \{0,1\}^{\Sigma_{\mathsf{T}1}} \\[0.5em] \text{⑧}\ \pi_{\mathsf{T}2} \in \{0,1\}^{\Sigma_{\mathsf{T}2}}. \end{cases}$$

## C  Tabular Experiment Details

For 4-player Kuhn poker with 13 ranks and 4-player Liars dice (coin) with 2 outcomes we used the original multiplayer game but had player 0 and player 2 be on one team and player 1 and player 3 be on the other. Then each player's utility is the average of their original utility and their partner's. This version of Kuhn poker has 566280 leaves and this version of Liar's coin has 4080 leaves.

## D  Details of Google Research Football Training

The Google Research Football (GRF)Kurach et al. (2020) environment is a simulation environment for real-world football games, where each game consists of 3000 steps. Players need to control one or more players to compete with the opposing team, and the rules are the same as in general football games.The team with the highest number of goals at the end of the match wins. To validate the effectiveness of our algorithm, we conducted training and evaluation of our algoritm on the full 5 vs. 5 game in GRF, where the goalkeeper is controlled by built-in AI, while the rest of the four players are controlled by our model.

**Feature Engineering and Action Space.** Given the default setting where every player has access to complete game information, we have meticulously crafted a 133-dimensional feature vector to

enhance the training process. The input layer of the network is structured as [133 × 128] to ensure effective feature representation and processing. Furthermore, the player's action space is defined using a default set of 19 discrete actions, encompassing fundamental moves such as movement, passing, and shooting. Consequently, the output layer of the network is configured as [128 × 19], allowing for the mapping of features to appropriate action selections.

**Reward Shaping.** Due to the sparsity of rewards in the GRF environment, it is crucial to design appropriate reward mechanisms. In addition to the rewards provided by the environment, we have designed specific reward components, including win reward, goal reward, yellow card reward, checkpoint reward and lost ball penalty, to expedite our training process and enhance its effectiveness. We adopt a consistent reward configuration across all our experiments.

**Training.** For each algorithm, our models were trained for 1500M steps on a cluster with 256 CPUs and 8 RTX3090 GPUs, using three different random seeds. The batch size for each GPU was set to 512 and The discount factor $\gamma$ was set to 0.9999. We employed the Adam optimizer with a learning rate of 0.00005.Specifically, for team PSRO-MM, to mitigate the increased training cost associated with incorporating excessive mix strategies, we implemented a strategy of introducing two mix strategies to the population in each round (We set two here, however, you can add more if you wish). Each mix strategy was constructed by randomly selecting pure strategies from past checkpoints and assigning them to individual players within the team. This approach allows for a controlled inclusion of mix strategies while managing the overall training overhead, enabling an efficient training process while benefiting from the incorporation of diverse strategies within the population. More hyper-parameters are summarized as bellow:

| Hyper-Parameters | Value |
| --- | --- |
| game length | 3000 |
| batch size | 512 |
| max clipped value loss | 0.2 |
| gradient clip norm | 10 |
| value loss | huber loss |
| discount factor $\gamma$ | 0.9999 |
| learning rate | 5e-4 |
| ppo update number | 5 |
| gain | 0.01 |
| GAE $\lambda$ | 0.95 |
| entropy coefficient | 0.001 |

Table 1: Hyper-parameters

**Evaluation Metrics.** Due to the complexity of the GRF, calculating the exploitability of a specific policy or Predictive Error (PE) for a certain population becomes challenging. Both metrics involve max or min operators, and approximating the best response can be significantly inaccurate in such a complex game. We use the relative population performance (RPP)Vinyals et al. (2019a) as the metric. RPP represents the game interaction results of two populations A and B under their meta-nash equilibrium. Besides, onsidering our objective of identifying robust policies with strong performance in real-world games, we assess the effectiveness by comparing the average goal differences, score and Elo between the policies obtained from different methods and the built-in bots with varying difficulty levels within the General Reinforcement Learning Framework (GRF). This evaluation allows us to gauge the capabilities of the policies in challenging game scenarios and their ability to outperform predefined bot opponents.

**Elo Calculation** Elo refers to the Elo rating system used to measure the relative skill levels of the algorithms. For the computation of Elo ratings, they was computed by making agents play against each other. Here is a detailed explanation of how they were computed:

- Assign an initial Elo rating (1000 in our setting) to each player.
- Determine the expected score of each player in a game. This is calculated as follows:
    - For player A: `expected score` $= \dfrac{1}{1 + 10^{\frac{(B-A)}{400}}}$

- For player B: `expected score` = $\frac{1}{1+10^{\frac{(A-B)}{400}}}$ where A and B are the current Elo ratings of the two players.

- Play the game and determine the actual score of each player.

- Update the Elo ratings of the two players based on the outcome of the game and the expected scores:
  - For player A: `new rating = old rating` + $K$ × `(actual score − expected score)`
  - For player B: `new rating = old rating` + $K$ × `(expected score − actual score)` where $K$ is a constant that determines the "weight" of the update (set as 10 in this case).

- Repeat the process for each game, using the updated Elo ratings from the previous game as the starting point for the next game.

**Additional Experiments.** Since each player can observe the entire game information in GRF, we conducted experiments involving feature engineering to create an imperfect information scenario in the main body. In this scenario, each player is only able to observe information related to teammates, the ball, and the three nearest opposing players, which means there is always a lack of information regarding two opposing players. However, we also conducted experiments with perfect information, as depicted in the figures. Remarkably, we observed that the same pattern held true in both imperfect and perfect information scenarios, with Team PSRO-MM outperforming Team PSRO, which in turn outperformed self-play.

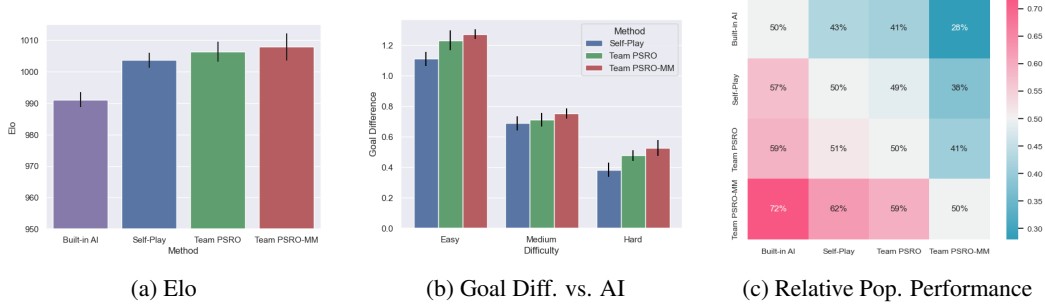

(a) Elo  (b) Goal Diff. vs. AI  (c) Relative Pop. Performance

Figure 3: Team PSRO Results on Google Research Football perfect information scenario. Both Team PSRO and Team PSRO-MM outperform self play, with Team PSRO-MM performing the best.

Additionally, we aimed to compare the performance of both Team PSRO and Team PSRO-MM as the training time steps increased, using the policy trained by self-play as the benchmark. Since draws can occur in football, win rate may not accurately reflect the policies' performance. Therefore, we used average scores as the evaluation metric, where the score for a single game is 1 for a win, 0.5 for a draw, and 0 for a loss. To better reflect the relative strength of each policy against the other, we multiplied the average scores by 100.

The results of the experiments are presented in Figure 4, with the dashed line representing the performance of the policy trained through self-play. We can see that both Team PSRO and Team PSRO-MM ultimately outperform self-play. Additionally, Team PSRO and Team PSRO-MM exhibit comparable performance in the initial stages of training. However, as the training timesteps progress, Team PSRO-MM exhibits superior performance, surpassing Team PSRO. This signifies that the integration of mixed strategies enhance the training effectiveness and robustness of the models. With an extended duration of training, Team PSRO-MM demonstrates a heightened ability to adapt to complex game environments, yielding higher-quality strategies.

Finally, we include additional comparisons with fictitous play (FP) and prioritized fictitous self play (PFSP). These methods learn a team best response to either the opponent average strategy (in FP) or a mixture of the average and the most recent (PFSP). We show in figure 5 that Team PSRO and Team PSRO-MM outperform these additional baselines as well.

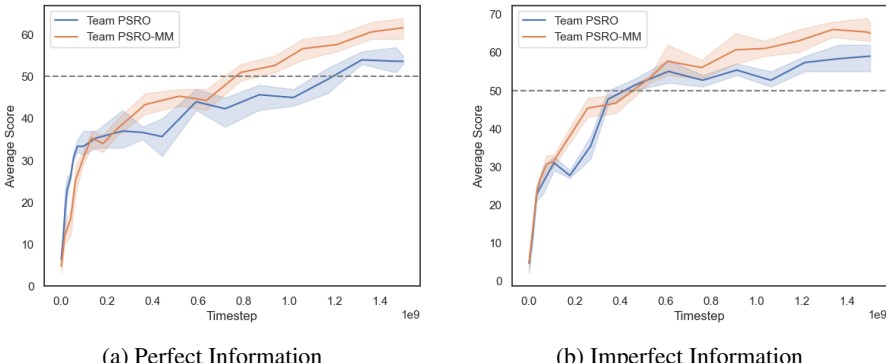

(a) Perfect Information         (b) Imperfect Information

Figure 4: Team PSRO results on both perfect and imperfect information scenarios of Google Research Football. Both Team PSRO and Team PSRO-MM outperform self-play, with Team PSRO-MM performing the best.

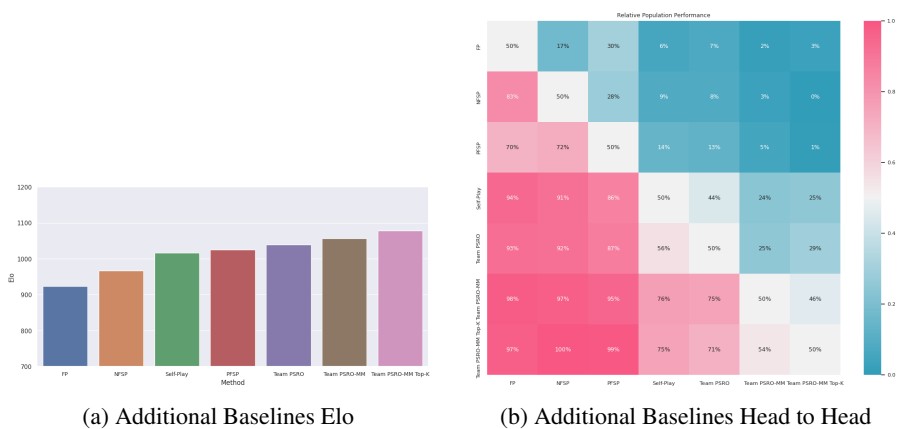

(a) Additional Baselines Elo         (b) Additional Baselines Head to Head

Figure 5: We include additional baselines of fictitious play and prioritized fictitious self play and show that Team PSRO and Team PSRO-MM are able to outperform all existing baselines.

