# OpenReview forum: "Team-PSRO for Learning Approximate TMECor in Large Team Games via Cooperative Reinforcement Learning"
_NeurIPS.cc/2023/Conference — NeurIPS 2023 poster_

### Official Review · Reviewer_eV3g · 2023-06-14

**Soundness:** 3 good
**Presentation:** 3 good
**Contribution:** 2 fair
**Rating:** 5
**Confidence:** 3

**Summary:**

This work addresses the problem of solving large-scale zero-sum two-team games. To solve this they explore extensions to the Double Oracle and Policy-Space Response Oracle algorithms that solve for a team-based equilibrium concept called TMECor. This is a straightforward extension of both of these algorithms that modifies the equilibrium concept and how policies are added to the empirical game. They evaluate their methods on the four-player variants of Kuhn Poker, Liar's Dice, and Google Football.

**Strengths:**

- Background clearly explains all of the necessary elements required.
- The subscripting used in notation throughout that uses both color and unique font (colorblind-friendly) helps keep keep concepts very clear and organized.
- This paper is honest and humble with how much of their work is building upon preexisting work.
- Include discussions of their results on both algorithmic iterations and walltime.


**Weaknesses:**

- Related work contains a lot of very tangential references. For example, the various PSRO extensions. Later in the paper a variety of significantly more related work is discussed, it would make sense for this to be in the related work section instead.
- L232: This reads like the authors are claiming using DRL as a BR oracle is a contribution of this work, when it is one of PSRO.
- L238-244: This is pretty challenging to read and is a single sentence, strongly recommend editing into individual statements.
- In the PSRO sections, the empirical game is treated as a restricted game. This is misleading and incorrect, because the payoffs are estimates.
- No baselines are considered in DO variants of their algorithms.
- Limited evaluation of the PSRO variants of the algorithms.
- Missing approximate exploitability curves of PSRO algorithms. Consider also performing a combined-game analysis.

**Questions:**

- L33: Why are you considering TMECor and what other alternatives did you consider and why not them?
- Could you please cite and define TMECor and its abbreviation?
- L56: The stratego paper explicitly says they use self-play. Could you please provide a citation that the in fact are using Pipeline PSRO?
- L123: I thought team games defined the players sharing payoffs. Why are they being separated here?
- L247: What is the "variant" of the algorithm? What specific changes are being made that are novel to this work and why are they necessary? If applicable, can you show gains from these changes?
- This work could be enhanced by including baseline method(s). For example, natural one is to consider that each teamplayer plays the same policy, where the policy parameters are a correlation device and their private information realizes different behaviors. Another option is to treat each team as a single agent and then an algorithm like QMIX to factor out per-player behavior. Without something akin to these baselines, it's hard to understand what benefit of the team-like treatment applied here.
- L292-293: It's not clear to me that this is true. Surely, if the first 1-n BR are exact and the last, n, iteration is an epsilon BR, then the final result will be an approx TMECor. However, the compounding errors of having epsilon BRs at every iteration makes it hard to say anything about convergence. Or are you trying to make a super weak claim about how approximate the TMECor is? If so, I suggest being more clear about this.
- L305: If the teammates are sharing policy parameters what is really the difference between Team-PSRO and Team-PSRO-MM? The strategy sets are the same?

**Limitations:**

Adequately discusses limitations with team double oracle (L220-225), and generally (L356-362).

---

> ### Author Rebuttal · Authors · 2023-08-10
>
> We would like to thank the reviewer for the detailed review and valuable feedback on our work. These insights have greatly helped us in identifying areas of improvement. We would like to address the concerns as follows:
>
> **Strengths**
>
> We're pleased that the reviewer appreciated our clear background explanation, unique notation, honest portrayal of our contributions, and the inclusion of discussions on algorithmic iterations and walltime.
>
> **Weaknesses**
>
> Related Work Organization: We agree that some references in the related work section could be reorganized. We will revisit this section and ensure that the most relevant works are appropriately placed and discussed.
>
> Lines 232, 238-244: We understand your concerns and will clarify our contributions and rephrase the complex sentence to enhance readability.
>
> Treatment of the Empirical Game: Your point about our phrasing of the empirical game as a restricted game is well-taken, and we will edit this description to avoid any confusion.
>
> Baselines in DO Variants: Thank you for this great suggestion. We have included additional experiments against fictitious team play [FCGS18]. As shown in the figure in our rebuttal pdf, our methods outperform fictitious team play. We believe that these additional experiments greatly improve the paper.
>
> Evaluation of PSRO Variants: We will work on providing more comprehensive evaluation details for the PSRO variants.
>
> Approximate Exploitability Curves: Our decision not to compute exploitability in Google Research Football (GRF) aligns with previous research, given the substantial computational resources required and the potential for high inaccuracy due to the environment's complexity and randomness.
>
> **Questions**
>
> Consideration of TMECor: When it comes to adversarial team games in which the players cannot secretly communicate, there are two main notions of equilibrium that are considered: TME and TMECor. TME corresponds to having the team members play independently, while TMECor allows for correlation between the team members. We believe there are several reasons why TMECor is the superior solution concept for the setting we consider. First, TMECor always guarantees a higher value for the team. Second, correlation between team members can be achieved easily, for example by communicating before the game starts, as in bridge. Finally, TMECor is known to be learnable, while TME is significantly harder, both in theory and in practice (for example [ZFS23] shows that TME is in a harder complexity class than TMECor). We discuss this in lines 164-170 but will include expanded discussion. We define TMECor in lines 128-132 (we will add a citation to the original paper too in the final version).
>
> Stratego Paper and Pipeline PSRO: Sorry for the confusion. The Pipeline PSRO paper achieves state-of-the-art performance in Barrage Stratego. We will edit the paper to reflect this fact.
>
> Team Games and Payoffs: Although different players can achieve different utility, the team utility is the sum of both players, so solving for TMECor can be viewed as a setting where both players have the same utility (the team utility). We will add discussion to make this more obvious.
>
> Novelty of Algorithm Variants: We will clarify the specific novel changes in the algorithm variant and demonstrate the necessity and gains from these changes.
>
> Inclusion of Baseline Methods: Regarding the suggestion of having each team player play the same policy: in fact, that is exactly what we do with MAPPO. Each team player has the same weights. But we don’t just have one policy, we have multiple in a population. We already include a self-play variant where there is just one policy, which I think is what is being referenced. As to including QMIX ablations, we believe that this is not the interesting aspect of our approach. There are multiple existing methods for cooperative deep RL, including MAPPO, QMIX, and many others. We run experiments with MAPPO because it has been shown to be the best-performing algorithm, but perhaps different cooperative RL algorithms would work better in different domains. However, we leave this ablation study to future work, as the main contribution of our paper is showing that cooperative RL can be combined with a PSRO-based method to find approximate TMECor.
>
> Convergence: Actually we still have convergence to epsilon-TMECor if all best responses are epsilon-BRs. To see this, at convergence, we have that the best response cannot exploit the opponent meta-Nash more than epsilon over the existing meta-Nash for both teams, so by definition the meta-Nash is an epsilon-TMECor. We will make this point more clear in the paper and can add a brief proposition and proof if the reviewer would like.
>
> L305: The team members share parameters in the joint BR. So every strategy in Team PSRO will share parameters. But in Team PSRO-MM, the policy for each player can come from different iterations, which have different parameters. We will make sure this is clear in the revised version.
>
> **Conclusion**
>
> We acknowledge your concerns and agree that our paper requires improvements. We believe that the planned revisions, as addressed above, will significantly enhance the quality of our work and respond to your critique. We hope that these clarifications and commitments to improvements might lead you to reconsider the rating.
>
>
> [FCGS18] Gabriele Farina, Andrea Celli, Nicola Gatti, and Tuomas Sandholm. "Ex ante coordination and collusion in zero-sum multi-player extensive-form games." In: Advances in Neural Information Processing Systems (NeurIPS), 2018.
>
> [ZFS23] Brian Hu Zhang, Gabriele Farina, and Tuomas Sandholm. "Team Belief DAG: A Concise Representation for Team-Correlated Game-Theoretic Decision Making.". In: International Conference In Machine Learning (ICML), 2023.
>
> [CG18] Celli, Andrea, and Nicola Gatti. "Computational results for extensive-form adversarial team games." In: AAAI Conference on Artificial Intelligence (AAAI), 2018.

---

> > ### Comment · Reviewer_eV3g · 2023-08-10
> >
> > Thank you for answering my questions and addressing the errors within the manuscript.
> >
> > To follow-up on the existing points of discussion on my review:
> > - Thank you for including the baseline, this definitely helps contextualize the results. I am still surprised to not see a treatment of DO using some form of team reduction. This would be a much closer comparison but is not a reason to block this work.
> > - I do think including additional ablations/comparisons for PSRO will better help the readers to better understand this work. However, contrary to my fellow reviewer MAu9, I disagree that it is unreasonable to request methods that "can be easily extended ... with some modification", because the modifications themselves warrant independent studies to verify their correctness.
> > - I would suggest adding the discussion about TME and TMECor within the final version of the manuscript. NeurIPS is not primarily a game theory audience so I think this would be very welcome.
> > - In regards to the baseline methods, I understand the MAPPO and self-play methods. My point was more to better understand what benefits were empirically gained for explicitly training these agents as a "team". I would argue that the viewing of common-interest agents as analogous to the decentralized control of a single agent is the most immediate and direct comparison. Therefore, by using it as a baseline, we better understand what is of import with treating them as individuals instead of as a collective single entity. In my opinion, this result, which is unreasonable to ask for in this review period, would be the most convincing piece of evidence to me in support of your claim.
> >
> > I still have some reservations about this work and whether the team aspect of it is any different than a decentralized single-agent problem, and some of the experimental things we've discussed. However, my fellow reviewers Gu1R and kqkC are happy with it, and I don't believe MAu9 raised any concerns necessitating the rejection of this work, so I am willing to change my score to a borderline/weak accept. However, I would neither fight for the acceptance nor rejection of this work.

---

> > > ### Author Response · Authors · 2023-08-14
> > > **Author Response**
> > >
> > > We sincerely appreciate your thoughtful follow-up and the time you've invested in evaluating our work. Your insights have been instrumental in identifying areas for improvement, and we are committed to addressing them in the final version of the manuscript. Below, we respond to your specific comments:
> > >
> > > **Baseline for DO Using Team Reduction:** What exactly do you mean by team-reduction? If you mean trying QMIX as an ablation, we would expect Team PSRO with QMIX to underperform Team PSRO with MAPPO since both QMIX and MAPPO are cooperative RL algorithms but MAPPO empirically performs better across a wide range of environments. We will certainly consider this aspect in future work and appreciate your understanding that it's not a reason to block the current submission.
> > >
> > > **Additional Ablations/Comparisons for PSRO:** We have added a novel algorithm, called Team PSRO-MM Top-K, where we select the top k strongest opponents (in this case k equals 4) after every iteration of evaluation and use these policies to get a mixed policy to add to the population. We have done additional experiments to compare our method with other baseline methods in GRF, including a deep RL version of fictitious team play [FCGS18] and PFSP [AlphaStar]. **We have included these experiments in the rebuttal pdf and show that the PSRO-MM Top-k outperforms all baselines, including the added baselines of fictitious team play and PFSP.**
> > >
> > > **Discussion about TME and TMECor:** Your suggestion to include a more detailed discussion about TME and TMECor is well-received. We recognize that NeurIPS has a diverse audience, and we will ensure that our final manuscript includes a comprehensive explanation that caters to readers with varying backgrounds in game theory.
> > >
> > > **Understanding the Benefits of Treating Agents as a Team:** We understand your interest in exploring the empirical gains of training agents as a team versus treating them as a decentralized single-agent problem. Since finding a joint best response is naturally a cooperative RL problem, we anticipate independent RL to perform worse than MAPPO, as has consistently been demonstrated across a wide range of cooperative RL environments. We will consider this direction in future research and appreciate your understanding of the limitations within the current review period.
> > >
> > > In conclusion, we have shown that finding TMECor in large games can be reduced to a cooperative RL problem. **While ablations on the type of cooperative RL algorithm used (MAPPO vs. QMIX vs. independent RL) are potentially interesting, the main point of our paper is that one can use whichever cooperative RL algorithm they choose.** We chose MAPPO due to its demonstrated empirical superiority compared to other algorithms, but one could choose a different cooperative RL algorithm and we would still expect our method to work. We have also included a new algorithm and new baselines for our deep RL experiments.
> > >
> > > We are grateful for your willingness to change your score to a borderline/weak accept. We understand that you have reservations, and we are committed to addressing them to the best of our ability in the final manuscript. Your feedback has been invaluable, and we believe that the planned revisions will enhance the quality and impact of our work. Thank you once again for your constructive feedback and consideration.

---

> > > > ### Comment · Reviewer_eV3g · 2023-08-15
> > > >
> > > > Thanks for the reply. I think I may have been unclear in my previous message, so I am attempting to clarify the points below:
> > > >
> > > > **Single-Agent vs Team:** Sorry if this was unclear, this point relates both "Baseline for DO Using Team Reduction" and "Understanding the Benefits of Treating Agents as a Team". Consider we are interested in solving a 2v2 game. A baseline method would be to apply to PSRO _directly_ on a reduced version of the game, where we abstract the teams away so the game is 1v1. This game could be solve directly and then its final solution could be factorized into independent player controllers (through eg QMIX). The factorized game can then be solved for TME/etc.. One could also consider a similar baseline where at each round of PSRO in the 1v1 game the policies are factored and used to inform game-reasoning of the unabstracted game at each iteration. The major point here being it's not clear to me how using the prior of TME/TMECor throughout game-solving is advantageous when compared to abstractions methods. The authors never claim that Team-PSRO is advantageous to alternative game representations, which is why I am not willing to block this work. However, I think how Team-PSRO compares to game abstraction dramatically effects my impression of the "impact" of the method, so I would need to understand this relationship to be truly excited about the method. I think this is also an angle where the authors could show larger benefits of their "mix and match" procedure.
> > > >
> > > > **Independent RL vs MAPPO:** I would disagree with the authors here, it's not clear to me how "independent RL [will] perform worse than MAPPO", when independent RL controls the full team. I think this claim might have been due to a misunderstanding of my previous messages. Does my previous bullet point help clarify my position?
> > > >
> > > > **Comparisons:** Sorry, it looks like there was a pretty severe typo in my first reply. I think it is unreasonable to request methods that "can be easily extended ... with some modification". This is why I think my fellow reviewer has not presented any valid reasons for rejecting this paper.

---

> > > > > ### Author Response · Authors · 2023-08-16
> > > > > **Author Response**
> > > > >
> > > > > Thank you for your continued engagement and for taking the time to clarify your points.
> > > > >
> > > > > **Single-Agent vs Team**: If we understand correctly, you're suggesting that we solve the game by treating each team as a single player (for example in bridge, team members can see each other's cards) and then extract a team strategy from that solution using QMIX. While this approach is indeed a potential baseline, we have reservations about its applicability to finding TMECor. For example, in bridge, the uncertainty regarding a partner's cards is a fundamental aspect of the game. An optimal strategy for a single player that has visibility into both sets of cards would bypass the need for signaling or any form of communication with the other team member, which is a core element of bridge. Such a strategy would not converge to TMECor, our solution concept of interest.
> > > > >
> > > > > Regarding your suggestion that "each round of PSRO in the 1v1 game, the policies are factored and then utilized to inform the game-reasoning of the unabstracted game at every iteration," could you elaborate further? We're not entirely clear on this point, though we believe our earlier response might still pertain to this recommendation.
> > > > >
> > > > > You say that we use the prior of TMECor instead of using abstraction. Would you please explain what you mean by using a prior of TMECor? We are *solving* for TMECor because TMECor is a natural solution concept in this setting. Our algorithm converges to TMECor, and the proposed abstraction algorithm, if we understand what you are proposing correctly, doesn’t. Hence, while we could consider this experiment in future work, we opted not to pursue it in the current study, anticipating its divergence from TMECor.
> > > > >
> > > > > **Independent RL vs MAPPO**: We apologize for any confusion earlier. Our reference to "independent RL" was in the context of learning a joint best response without information sharing (i.e., without seeing each other's cards). In such a scenario, learning a joint best response becomes a cooperative RL problem, and it would be prudent to employ the best cooperative RL algorithm available. However, if the suggestion is to treat the team as a single player (where team members can see each other's cards), then the problem indeed transforms into a single-agent RL problem.
> > > > >
> > > > > **Comparisons**: We're grateful for your understanding regarding the comparisons. We recognize the challenges in extending methods with significant modifications, and we appreciate your stance that it's unreasonable to request such methods in the review process.
> > > > >
> > > > > Please let us know if we misunderstood anthing. We hope that our clarifications address your concerns, and we look forward to your continued feedback.

---

> > > > > > ### Comment · Reviewer_eV3g · 2023-08-18
> > > > > >
> > > > > > I think the common thread of confusion is the implicit assumption that treating a team as a single-agent somehow necessarily makes it so that there is "one policy" and that all agents have access to "all information". This is not requisite. You can treat a team as a single-agent with a structured action space as I expect you do with MAPPO. I believe the core of this confusion is that I suggested QMIX as one way to perform such an abstraction, which factors a policy. However, note that even for QMIX, the key is to factor the policies in a way that they do not depend on other player's private information. Again, policies may be trained with private information so long as they do not depend on it during evaluation. This typically affords more efficient learning.
> > > > > >
> > > > > > I apologies that "prior" may have been too loaded of a term to adequately convey my point. My goal is to understand how much of a contribution this work is to the literature so that I can quickly accept the paper. The problem I come into is _if_ the main methodological change is to swap the MSS with a TMECor solver then the novelty is marginal (I realize you also included additional novelty with your "M&M"). The _simplest_ way I could think of to compare what benefit this choice has, is to just run PSRO say with NE solver, and then compute a TMECor at the end. This would give you some measure as to how much biasing your strategy exploration method with TMECor actually mattered.
> > > > > >
> > > > > > This second point is muddled with my first point, because I don't see how common-interest agents should be treated different than a single-agent problem in centralized and simulated learning problem statements. This furthers my need to understand what actually is helping.
> > > > > >
> > > > > > Does this help clarify my position?

---

> > > > > > > ### Author Response · Authors · 2023-08-19
> > > > > > > **Author Response**
> > > > > > >
> > > > > > > Thank you for clarifying. We believe you are proposing two different methods.
> > > > > > >
> > > > > > > The first is to use QMIX instead of MAPPO, but keep everything else in our algorithm the same. This ablation would be interesting, and would potentially improve performance, but we did not have time to run it during the rebuttal period. However, our method is agnostic to the cooperative RL algorithm used, so this proposal is best seen as an ablation of one aspect of our algorithm.
> > > > > > >
> > > > > > > The second proposal, if we understand correctly, is as follows: every iteration, train a joint best response (as we currently do), but change the meta-strategy-solver to be a NE on the 4-player game. Then, at some point, compute a TMECor on the meta-game. While this approach is interesting, this approach will not be guaranteed to converge to TMECor. This is because there could be certain scenarios where the joint best responses to the 4-player NE will be the same from one iteration to the next, but the TMECor of the meta game will not be the TMECor of the full game. Our method fixes this problem by computing a TMECor on the meta-game every iteration, and will converge to a TMECor, as shown theoretically and empirically in our paper. Furthermore, finding a NE in general-sum games is PPAD hard, so the proposed method will not scale as the meta-game gets larger.
> > > > > > >
> > > > > > > We understand that this nuanced point was not sufficiently conveyed in our paper, and we will work to make this point more clear. Although we were not able to run new experiments with the second proposed method (of computing NE in the meta-game), we would be happy to include tabular results of this method for the camera-ready version and are confident that we can show that there exist games where the new proposed method will not converge to TMECor.
> > > > > > >
> > > > > > > Did we correctly understand the new proposed methods? Or perhaps were you proposing that in the second proposed method the best responses aren't joint best responses but independent best responses? If so, the same arguments hold for why that method will not converge to TMECor as well.

---

### Official Review · Reviewer_Gu1R · 2023-06-27

**Soundness:** 3 good
**Presentation:** 3 good
**Contribution:** 2 fair
**Rating:** 6
**Confidence:** 3

**Summary:**

This paper proposes two algorithms, “Team PSRO” and “Team PSRO Mix-and-Match” for zero-sum two-team games. Team-PSRO is guaranteed to converge to a TMECor. The algorithms extend PSRO to zero-sum two-team games. Team-PSRO Mix-and-Match is an improved version of Team-PSRO with better population policies. The experimental results show the convergence of Team DO in Kuhn poker and Liar’s dice, and Team PSRO beats self-play in the Google Research Football environment.


**Strengths:**

- Extend PSRO to “Team PSRO”, and propose “Mix-and-Match” Team PSRO.
- Team-PSRO is guaranteed to converge to a TMECor.
- The experimental results show the convergence of Team DO in Kuhn poker and Liar’s dice, and Team PSRO beats self-play in the Google Research Football environment.


**Weaknesses:**

- The description about Team DO-MM and Team PSRO-MM is unclear. For example, the function P is not clearly described, so it becomes unconvincing for the success of the new method.
- It is unclear about how NE is derived exactly in TEAM-PESO when implementing it. It would be also helpful if an anonymous github is given or in the supplementary attachment.


**Questions:**

The citations are written in a very confusing way. I believe you use a wrong latex command.


**Limitations:**

N.A.

---

> ### Author Rebuttal · Authors · 2023-08-10
>
> We thank the reviewer for their careful examination of our paper and their valuable comments. In response to their concerns, we provide the following explanations and planned corrections:
>
> **Weaknesses**
>
> Unclear Description about Team DO-MM and Team PSRO-MM: P is defined in line 262. We will move this definition to the notation section to make it clearer.  We will also work on improving the text to better elucidate the concepts and methods involved in Team DO-MM and Team PSRO-MM.
>
> Deriving NE: The paper indeed missed a detailed description of how NE (Nash Equilibrium) is computed in TEAM-PSRO. The short answer is that we solve it exactly using an LP. We will add the required details in the revised version, making sure it's clear and understandable.
>
> Code Availability: Your suggestion about providing an anonymous GitHub link or supplementary attachment with the code is valuable. We will try to provide an anonymized repository for the camera-ready version.
>
>
> **Questions**
>
> Confusing Citations: We apologize for the confusion caused by the way citations are written. We suspect a technical issue in the LaTeX formatting. We'll correct this in the revision, ensuring that the citations follow a consistent and standard format.
>
> **Conclusion**
>
> We are committed to addressing all the points you have raised and believe that these revisions will substantially enhance the clarity and completeness of the paper. We hope that our response assures you of our determination to deliver a high-quality paper and that you might consider a higher rating. Thank you once again for your thoughtful review, and we look forward to your further feedback.

---

### Official Review · Reviewer_kqkC · 2023-07-03

**Soundness:** 3 good
**Presentation:** 4 excellent
**Contribution:** 3 good
**Rating:** 6
**Confidence:** 4

**Summary:**

The paper presents “game-theoretic” reinforcement learning methods for playing zero-sum games (between teams of players).  A theoretical claim (proof is in the supplementary material) is made about convergence of the base tabular methods to the TMEcor solution concept, and empirical results compare the methods to self play RL in the “Google Research Football” domain.


**Strengths:**

1. By and the large, the paper is presented very well.  Concepts are explained well, the paper is very polished.  The derivation and explanation of methods appear to be sound.

2. The topic seems important, with well-defined limitations

3. The results appear to show good (albeit incremental) performance, though see statements in “weaknesses” for clarification

4. The enhanced methods are simple

Overall, this seems to be a useful paper.


**Weaknesses:**

1. The results need better explanation and analysis, particularly surrounding the presentation of Figure 2.  Not enough information is supplied.  I have the following questions: (1) Were appropriate statistical tests run to evaluate statistical significance?  If so, they should be reported.  If not, claims of being better are unsubstantiated.  (2) What do the error bars in Figure 2a-b represent?  (3) What is Elo (Figure 2a)?  (4) What is the built-in AI against which the algorithms were paired?  Why only the comparisons when paired with built-in AI?  What about results when self play RL and PSRO-MM were the opponents?  (5) What about RL algorithms trained against a variety of opponents instead of just self play?

I think these questions are easily addressable, and understand the difficulty of fitting everything into a short conference paper.  It remains to be seen whether the clarifications would favor the proposed techniques or not.

2. The paper has a couple of unsubstantiated claims that, in my opinion, need to be removed or altered:

A) The paper claims to “introduce *the first* game-theoretic techniques for two-team games” (emphasis added).  The paper does not prove this to be true (it is doubtful this could be proven), and often such claims are not true.  The paper later re-states the claim with the caveat of it being the first “to their knowledge,” which is better but still unsubstantiated.  In general, the claim about being “the first” has little to no scientific value and causes the reader to focus on whether it is indeed “the first” rather than the contribution of the paper.  I’d recommend the statements be removed altogether.

- Second, the paper claims (intro to section 4) that tabular methods will not scale to large games.  That may or may not be true (seems like there are ways it could be done, and perhaps very well, if one looks at it from a different paradigm).  Regardless, the paper does not back up the claim.  I think the statement should be softened or at least the opinion be given better context.


**Questions:**

The results need better explanation and analysis, particularly surrounding the presentation of Figure 2.  Not enough information is supplied.  I have the following questions: (1) Were appropriate statistical tests run to evaluate statistical significance?  If so, they should be reported.  If not, claims of being better are unsubstantiated.  (2) What do the error bars in Figure 2a-b represent?  (3) What is Elo (Figure 2a)?  (4) What is the built-in AI against which the algorithms were paired?  Why only the comparisons when paired with the built-in AI?  What about results when self play RL and PSRO-MM were the opponents?  What about RL algorithms trained against a variety of opponents instead of just self play?

I'm potentially inclined to change my review based on the answers to these questions.


**Limitations:**

Yes, I think the paper does a good job with this.

---

> ### Author Rebuttal · Authors · 2023-08-10
>
> We greatly appreciate the reviewer's time and detailed feedback. We aim to address all the concerns mentioned:
>
> **Results Explanation and Analysis (Figure 2)**
>
> Error Bars and Statistical Significance: We acknowledge that the information about statistical tests was not included in the main text. Indeed, we performed statistical tests for the comparisons, and the results were statistically significant. This will be clarified in the revised version. The training details and additional experiments can be found in the appendix. For Figure 2a-b, we made the agents trained by different algorithms play against built-in AI with different difficulties and compared the goal difference. To avoid the influence of randomness, we ran three seeds for each experiment, the error bars in Figure 2a-b means the standard deviation. This comparative approach has also been employed in PSRO w. RD and BD[1] as a major evaluation method.
>
>
> Elo Explanation: Elo (Figure 2a) refers to the Elo rating system used to measure the relative skill levels of the algorithms.  For the computation of Elo ratings, they was computed by making agents play against each other. Here is a detailed explanation of how they were computed:
> - Assign an initial Elo rating (1000 in our setting) to each player.
> - Determine the expected score of each player in a game. This is calculated as follows:
> - For player A: expected score = 1 / (1 + 10^((B - A) / 400))
> - For player B: expected score = 1 / (1 + 10^((A - B) / 400)) where A and B are the current Elo ratings of the two players.
> - Play the game and determine the actual score of each player.
> - Update the Elo ratings of the two players based on the outcome of the game and the expected scores:
> - For player A: new rating = old rating + K * (actual score - expected score)
> - For player B: new rating = old rating + K * (expected score - actual score) where K is a constant that determines the "weight" of the update (set as 10 in this case)
> - Repeat the process for each game, using the updated Elo ratings from the previous game as the starting point for the next game.
> We will add an explanation of these details in the revised version.
>
> Explanation of Built-in AI and Comparison to other Methods: GRF incorporates  built-in AI agent and allows for difficulty adjustments. Consequently, in GRF-related experiments, the built-in AI is frequently employed as a benchmark to evaluate the performance of trained agents, such as PSRO w.RD and BD[1], TiKick[2], etc. We use relative population performance to evaluate the performance of different populations. In the appendix, in Figure-4, by using the agent trained by Self-play as the benchmark, we also compare the performance of both Team PSRO and Team PSRO-MM as the training time steps increased.
> New Self-Play Variant: While we haven’t seen the suggested approach of RL trained against a variety of opponents in the literature, and we don’t predict that such a method would have game-theoretic guarantees, this is an interesting suggestion for future work.
>
>
> [1] Liu, X., Jia, H., Wen, Y., Hu, Y., Chen, Y., Fan, C., ... & Yang, Y. (2021). Towards unifying behavioral and response diversity for open-ended learning in zero-sum games. Advances in Neural Information Processing Systems, 34, 941-952.
> [2] Huang, S., Chen, W., Zhang, L., Xu, S., Li, Z., Zhu, F., ... & Zhu, J. (2021). TiKick: towards playing multi-agent football full games from single-agent demonstrations. arXiv preprint arXiv:2110.04507.
>
>
> **Unsubstantiated Claims**
>
> First Scalable Game-Theoretic Techniques Claim: We do not claim to be the first game theoretic technique for two-team games. That is clearly not true based on the literature we cite in the paper. Instead, we claim to be the first **scalable** game-theoretic technique for two-team games. The current most scalable method for two-team games is Zhang et al. [ZFS23], which is a tabular method that clearly will not scale to google research football. We will soften the language a bit by claiming that *to our knowledge* we are the first scalable game-theoretic technique for two-team zero-sum games.
>
> [ZFS23] Brian Hu Zhang, Gabriele Farina, and Tuomas Sandholm. "Team Belief DAG: A Concise Representation for Team-Correlated Game-Theoretic Decision Making.". In: International Conference In Machine Learning (ICML), 2023.
>
>
> **Answers to Questions**
>
> We believe that the answers provided above address the questions raised by the reviewer.
>
> **Conclusion**
>
> We value the constructive feedback provided, and we believe that with the planned revisions, the paper will be strengthened significantly. The questions and concerns raised are indeed addressable, and we are committed to making the necessary changes to clarify all aspects.
>
> We hope that these explanations and our commitment to revise the paper accordingly will lead the reviewer to reconsider the rating. Thank you once again for the thoughtful review, and we look forward to your feedback on our responses.

---

> > ### Comment · Reviewer_kqkC · 2023-08-12
> >
> > Thanks for the response.  I'll keep my score as it stands.
> >
> > Notes:
> > 1.  Without seeing the results of the statistical tests, it is difficult for me to comment on them, though good that they are "statistically significant."
> >
> > 2.  I still think it would be better to not try to claim to be the first, even if claims are softened.  The community learns not by racing to the finish line, but by understanding the issues.  Subjective claims of being first divert our attention away from the issues.

---

> > > ### Author Response · Authors · 2023-08-14
> > > **Author Response**
> > >
> > > Thank you once again for your thoughtful comments and for maintaining your score. We appreciate your engagement with our work and your constructive feedback. We would like to respond to your notes:
> > >
> > > Statistical Tests: We understand your concern about not being able to comment on the statistical tests without seeing the results. In the final version of the paper, we will include the details of the statistical tests, ensuring that they are transparent and accessible to readers. We believe this will provide the necessary context and validation for our claims.
> > >
> > > Claim of Being the First: We acknowledge your perspective on the claim of being the first, even if softened. We agree that the focus should be on understanding the issues and contributing to the community's knowledge rather than racing to be the first. In light of your feedback, we will remove the claim altogether and concentrate on articulating the novelty and value of our approach without any comparison to the timing of other works.
> > >
> > > We believe that these changes will align with your suggestions and further improve the quality of our paper. We are committed to making these revisions in the final manuscript.
> > >
> > > Once again, we express our sincere gratitude for your time, effort, and valuable insights. Your feedback has been instrumental in guiding our revisions, and we look forward to incorporating your suggestions.

---

### Official Review · Reviewer_MAu9 · 2023-07-05

**Soundness:** 2 fair
**Presentation:** 3 good
**Contribution:** 2 fair
**Rating:** 4
**Confidence:** 4

**Summary:**

This work aims to find TMECor in two-team zero-sum games. They extend PSRO from two-player games to two-team games and proposed Team-PSRO which is guaranteed to converge to a TMECor. They further proposed Team-PSRO Mix-and Match which generates more joint policies by mixing individual policy from different PSRO iterations. They evaluate the proposed algorithms on Google Research Football and achieved better results than self-play.

**Strengths:**

This paper is clearly written and easy to follow. The use of symbols and definitions in the theory part is in accordance with the standard notations.


**Weaknesses:**

The novelty of the proposed algorithms is marginal, and more comparisons with closely related work are needed in the experiment. Please see below for detailed discussions.

1. Lack of novelty. Team-PSRO simply applies PSRO to two-team zero-sum games by learning a joint best response for the whole team, which has no difference from PSRO for two-player games and has been discussed in recent works like [4]. Team-PSRO-MM further proposes to mix individual policies in different iterations to generate more joint policies. However, mixing all individual policies will produce exponentially many joint policies and requires a lot more computation to get the payoff table. This paper simply mixes all policies or randomly samples policies to mix. It would help improve the novelty of this paper if the author can design a better way to smartly mix-and-match to produce joint policies that are most useful.
2. Need more baselines in experiments. For small games like 4-player Kuhn poker and Liar's dice, methods like NFSP [1], CFR [2] should be added as baseline. Though these methods are designed for two-player zero-sum games, they can be easily extend to two-team zero-sum games with some modification. For larger games like GRF, some closely related work like PSRO w. BD and RD [3], FXP [4] should be added as baseline. These methods also build on PSRO and report strong results in GRF full games. It is also straightforward to use NFSP in GRF.

[1] "Fictitious Self-Play in Extensive-Form Games." Heinrich, Johannes, Marc Lanctot, and David Silver.

[2] "Deep counterfactual regret minimization." Noam Brown, et al.

[3] "Unifying Behavioral and Response Diversity for Open-ended Learning in Zero-sum Games." Xiangyu Liu, et al.

[4] "Fictitious Cross-Play: Learning Global Nash Equilibrium in Mixed Cooperative-Competitive Games." Zelai Xu, et al.

**Questions:**

1. What is the difference between Team-PSRO and using PSRO by regarding the whole team as a single agent.
2. If there are n players in one team, and team-PSRO has run k iterations, then the number of joint policies produced by mix-and-match would be $k^n$, and it would require a lot of rollouts to complete the payoff matrix. Did the author produce all the joint policies in the experiments? Is there a better way to produce part of the joint policies that are most useful for training?
3. What is the rule of 4-player Kuhn poker and how many players are there in Liar's dice? These two games are originally two-player games and there is no description about how they are modified into two-team games.
4. In the experiment of GRF, how many iterations are trained for Team PSRO and Team PSRO-MM？Does the population start from random policy?
5. Comparison with NFSP and CFR in 4-player Kuhn poker and Liar's dice.
6. Comparison with PSRO w. BD and RD, FXP, NFSP in GRF.

**Limitations:**

The authors have discussed the limitations.

---

> ### Author Rebuttal · Authors · 2023-08-10
>
> We would like to express our gratitude to the reviewer for taking the time to assess our work and providing valuable insights. We would like to clarify certain aspects of our research that may not have been fully grasped, along with a plan to address the constructive suggestions.
>
> **Lack of Novelty**
> While it is true that our approach leverages the principles of PSRO, there are significant differences between applying PSRO to two-player games and two-team zero-sum games. The transition from the individual to team-based setting introduces complexities that need to be taken into account when designing algorithms. One consideration in the setting we study in this paper is that the players on the same team are not allowed to communicate with one another once the game has started.  **A direct consequence is that methods like CFR are not applicable, unlike what is stated in the review in Weakness 2 and Question 5.**
>
> We agree that a more sophisticated approach to mixing and matching could further improve performance. For example, we can select the top k strongest opponents (in this case k equals 4) after every iteration of evaluation and use these policies to get a mixed policy. Then we can add this policy to the population. We call this variant as Team PSRO-MM Top-K, and we have done additional experiments to compare our method with other baseline methods, including a deep RL version of fictitious team play [FCGS18] and PFSP [AlphaStar] (For PSRO w. BD and RD, it may take a lot of time to implement and train because they didn't reveal their code for GRF experiments.) **We have included these experiments in the rebuttal pdf and show that the PSRO-MM Top-k outperforms all other baselines.**
>
> Thank you for the reference to Xu et al. That paper indeed proposes a similar algorithm to Team-PSRO but it came out after the NeurIPS deadline so we were not able to cite it for the submission. We will be sure to reference it as contemporary work in the camera-ready version. However, in that paper the authors do not make a connection to TMECor and do not propose the idea of Team-PSRO-MM.
>
> Need for More Baselines in Experiments: We concur with the reviewer's observation that more baselines in experiments would provide a more comprehensive evaluation. **However, the reviewer’s suggestions for including methods like NFSP, CFR, and replicator dynamics do not make sense in our setting (see above).** NFSP, CFR, and replicator dynamics are all algorithms for two-player zero-sum games with perfect recall. They do not apply to two-team zero-sum games. **Although the suggested algorithms do not make sense in our setting, we have included new results with Fictitious Team Play [FCGS18]. As shown in the figure in our rebuttal pdf, we find that our methods outperform this baseline. For deep RL experiments, as mentioned above, we include baselines of deep RL versions of fictitious team play and PFSP, and show that we outperform them.**
>
> **Clarification on Specific Questions**
>
> Difference between Team-PSRO and single agent PSRO: Our setting considers games where team members cannot communicate during the game. As a result, Team-PSRO learns joint best responses for the team, taking into consideration the fact that team members cannot communicate. This differs significantly from considering the whole team as a single agent, which is equivalent to two-player zero-sum games with perfect recall. We study two-team zero-sum games which are equivalent to two-player zero-sum games with imperfect recall. In this setting, Team-PSRO converges to TMECor, instead of Nash as in two-player zero-sum games. We will add more discussion of this in the paper, although this is fairly well-known background information in the literature.
>
> Concerns about joint policies in Team-PSRO: We acknowledge the computational challenges related to the mix-and-match strategy, and we will expand on the methods used in our experiments to manage these complexities. Your suggestion for a more strategic approach is well-received, and we have included a new variant called PSRO-MM Top-k which outperforms all other methods. However, for team games with two players per team, which capture many domains of interest such as bridge, the number of joint policies only scales quadratically. As shown in our paper, Team-PSRO-MM seems to perform well in practice and is a valuable contribution as-is.
>
> Descriptions of 4-player Kuhn poker and Liar's dice: Your point is valid; we will provide detailed information on the modifications made to these games to adapt them to two-team formats in the revised version of the paper. These games are standard in the literature, for example in [FCGS18].
>
> Experiment details: Each policy was trained for 1.5×10e9 iterations. Since the random policy doesn't know how to pass or shoot, which makes it hard to train(especially in imperfect information scenarios), we first used RL to train a random policy against build-in AI with easy difficulty, and stopped training when win rate reaches 40%. The aim of it is to enable the agent to learn basic behaviors (especially shooting). Then we used Self-play and Team-PSRO to train the pretrained model. Specifics about the number of iterations and the starting policies for Team-PSRO and Team-PSRO-MM will be included to enhance clarity.
>
>
> Comparisons with other models: As noted above, NFSP and CFR are not valid algorithms for our setting, but fictitious team-play [FCGS18] is. We include additional experiments benchmarking against fictitious team play and show that our methods outperform fictitious team play.
>
> [FCGS18] Gabriele Farina, Andrea Celli, Nicola Gatti, and Tuomas Sandholm. "Ex ante coordination and collusion in zero-sum multi-player extensive-form games." In: Advances in Neural Information Processing Systems (NeurIPS), 2018.

---

> > ### Comment · Reviewer_MAu9 · 2023-08-14
> >
> > Thank you for your response and clarification. Some of my concerns have been addressed and here is a follow-up discussion.
> >
> > * Mix-and-match complexity: I agree with the authors that this is not a problem in games like Bridge, but it is a practical issue in GRF with 11 agents in each team. I'm glad to see the new variant called Team PSRO-MM Top-K and it is an empirical way to solve the problem.
> > * Description and Baselines for 4-player Kuhn poker and Liar's dice: the description of the modified team game would help the reader understand what method is applicable to these games, and I'm glad that the authors have taken my point. My previous concern is that these experiments only give results of the proposed algorithm and lack existing methods for comparison. The new result of fictitious team play serves as a good baseline.
> > * Baselines for deep RL experiments: I believe methods like NFSP are applicable in GRF and the needed change is to use cooperative RL instead of single-agent RL. However, as the authors have included new baselines like PFSP and fictitious team play, I think not adding my suggested baselines is acceptable.
> >
> > I appreciate the authors' effort to solve my questions above, but my main concern is still the novelty of the team DO/PSRO algorithm. My reasons are below.
> > 1. Minimal changes compared to DO/PSRO: if my understanding is correct, the only difference between team PSRO and PSRO is to use cooperative RL instead of single-agent RL for two-team zero-sum games (or to get joint BR instead of BR in team DO). This is a naive extension of PSRO and there is no new problem in doing so. In addition, the same algorithm has been used as a baseline in existing work like PSRO w. BD and RD, which makes me think the team DO/PSRO algorithm is of limited novelty.
> > 2. The only theory result (Proposition 1) is a direct corollary of a well-known theorem. In Theorem 1 of the double oracle paper [1], it is proved that DO converges to an NE. Because team DO simply replaces best responses with joint best responses, and the solution concept is TMECor instead of NE, the proof of Theorem 1 in [1] can be directly used for Proposition 1 and the proof in Appendix A does follow the same argument. This makes the only theory result of this paper not informative.
> >
> > Based on these two reasons, I think the first contribution described in L68, "We show that a straightforward extension of PSRO to team games converges to TMECor", is marginal. I would like to raise my rating to 4 based on the current discussions. And I'm willing to further raise the rating if my concern about the novelty is fully addressed.
> >
> > [1] McMahan, H. Brendan, Geoffrey J. Gordon, and Avrim Blum. "Planning in the presence of cost functions controlled by an adversary." Proceedings of the 20th International Conference on Machine Learning (ICML-03). 2003.

---

> > > ### Author Response · Authors · 2023-08-14
> > > **Author Response**
> > >
> > > Thank you once again for your thoughtful comments and continued engagement with our work. We appreciate your acknowledgment of our efforts to address your concerns, and we would like to further clarify the points you raised regarding the novelty of our approach.
> > >
> > > **Mix-and-Match Complexity**: We are pleased that you find our new variant, Team PSRO-MM Top-K, a satisfactory solution to the complexity issue in games like GRF.
> > >
> > > **Description and Baselines for 4-player Kuhn Poker and Liar's Dice**: We are glad that you agree with our inclusion of fictitious team play as a baseline and the detailed description of the modified team games. We believe these additions will provide readers with a clearer understanding of our methodology and its comparative performance.
> > >
> > > **Baselines for Deep RL Experiments**: Thank you for recognizing our efforts to include relevant baselines like PFSP and fictitious team play. We believe these comparisons provide a robust evaluation of our approach.
> > >
> > > **Concerns About Novelty**:
> > >
> > > **As we mentioned earlier, Xu et al. was published after the NeurIPS deadline, so it should not be used to argue against the novelty of our approach.** We acknowledge in the paper that Team-PSRO isn’t a radically new algorithm, but it forms the foundation of our other algorithms. Specifically, we claim our first contribution is showing that “*a straightforward extension of PSRO* to team games converges to TMECor.” The fact that we are the first to show this convergence is itself a meaningful contribution, regardless of whether it seems obvious in hindsight. **Most importantly, we introduce two additional algorithms: Team PSRO-MM and Team PSRO-MM Top-K.** Both of these algorithms represent substantial extensions and innovations compared to the two-player zero-sum PSRO algorithm. They outperform Team-PSRO and achieve state-of-the art performance on the domains we test.
> > >
> > > We acknowledge that Proposition 1 follows a similar argument to Theorem 1 in [1]. However, the adaptation of this theorem to the context of TMECor in two-team zero-sum games is a meaningful contribution. While the proof may follow a similar structure, the application to a new domain and the demonstration of convergence to TMECor are valuable insights that extend the existing understanding of these algorithms.
> > >
> > > **Conclusion**
> > > In conclusion, we believe that the challenges of two-team zero-sum games, the innovative solutions we have developed, and the empirical success of our approach collectively contribute to the originality and significance of our work. We hope that this response further clarifies our position and addresses your concerns. We are committed to making any additional revisions necessary to ensure that our contributions are fully understood and appreciated.

---

> > > > ### Comment · Reviewer_MAu9 · 2023-08-19
> > > >
> > > > Thanks very much for your reply. I would like to respectfully point out that, in my previous reply, the existing work that has already used team-PSRO as a baseline is PSRO w. BD and RD [1], not the work by Xu et al. **PSRO w. BD and RD is published in NeurIPS 2021, well ahead of this year's deadline, and should definitely be considered.** In addition, the authors do not provide any non-trivial difference between the proof of their theoretical result and the proof in DO/PSRO. In fact, I think previous work did not discuss this convergence because it is so obvious and is barely a theoretical contribution.
> > > >
> > > > Therefore, I'm still deeply concerned about the novelty of team DO and PSRO, and the second contribution (team PSRO-MM) alone does not meet NeurIPS' standards for acceptance. I will retain my current rating.
> > > >
> > > > [1] Liu, Xiangyu, et al. "Towards unifying behavioral and response diversity for open-ended learning in zero-sum games." Advances in Neural Information Processing Systems 34 (2021): 941-952.

---

> > > > > ### Author Response · Authors · 2023-08-19
> > > > > **Author Response**
> > > > >
> > > > > Thank you for your continued feedback. We'd like to directly address your concerns:
> > > > >
> > > > > **Liu et al. do not propose or experiment with team-PSRO.** Their work does not connect to TMECor because their method wouldn't converge to it. In Algorithm 1 of Liu et al., every iteration, a Nash equilibrium is computed on the n-player meta-game. **In two-team games, as described in detail in our paper, TMECor is not the same as finding a Nash equilibrium on the full game, which is what their method does.** We propose Team PSRO, which again is not equivalent to any version of Algorithm 1 in Liu et al. (i.e. even getting rid of diversity), and our method is guaranteed to converge to TMECor. This is a very significant distinction. **Liu et al.'s experiments are solely on two-player zero-sum games, not on two-team games as in our work.** Their GRF version has one player controlling the entire team, while ours has multiple team members with distinct information.
> > > > >
> > > > > **In summary, the reviewer's comparison between our work and Liu et al.'s is based on a misunderstanding. Their baseline and proposed methods are not equivalent to Team-PSRO and don't converge to TMECor. Our method does converge to TMECor, and we've provided both theoretical and empirical evidence for this.**

---

> > > > > > ### Comment · Reviewer_MAu9 · 2023-08-20
> > > > > >
> > > > > > Thank you for your reply. I respectfully disagree with the authors that "Liu et al.'s experiments are solely on two-player zero-sum games" and "Their GRF version has one player controlling the entire team". Their experiment considers the 11v11 full game which is a two-team game, and if they use one agent to control all the 11 players (each has 19 actions), the action space of this single agent would be $19^{11} > 10^{14}$, which is impractical. In addition, Table 5 in their appendix show that their policy network input contains information of the active player, which indicates that the policy controls a single player, not the entire team. The only possibility that they are treating this as a two-player game is that they only control 1 of the 11 players and let the built-in bot control the rest players, but I don't find any evidence in their paper to support this.
> > > > > >
> > > > > > Even putting our disagreement on Liu et al. aside, I am still not convinced by the authors' argument regarding the two concerns in my first reply, i.e. "minimal changes compared to DO/PSRO", and "the only theory result is a direct corollary of a well-known theorem". However, I do believe that both I and the authors have made our points clear in the previous discussion and I would like to thank the authors for their effort. I will keep my current score as it stands.

---

> > > > > > > ### Author Response · Authors · 2023-08-21
> > > > > > > **Author Response**
> > > > > > >
> > > > > > > Thank you for the continued dialogue. We'd like to clarify a few points regarding the comparison with Liu et al.'s work:
> > > > > > >
> > > > > > > **GRF Game Setting**: Although not explicitly mentioned in their paper, upon direct communication with the authors, they confirmed their use of the single-player GRF setting. In this mode, like you guessed, only the active player is user-controlled, while the rest are managed by the built-in AI. As further evidence, some of the authors of that paper secured second place in a Kaggle GRF competition [(link)](https://www.kaggle.com/competitions/google-football/discussion/202977), which specifically employed the single-player GRF setting.
> > > > > > >
> > > > > > > **Team-PSRO and TMECor**: As previously mentioned, Liu et al. neither propose nor experiment with team-PSRO. It's crucial to understand that in two-team games, as elaborated in our paper, achieving TMECor is distinct from computing a Nash equilibrium for the entire game, which is the objective of their method.
> > > > > > >
> > > > > > > We trust this clarifies the distinctions between our work and Liu et al.'s.
> > > > > > >
> > > > > > > As to the simplicity of our enhanced method, we agree with the other reviewers that this is a strength and not a weakness. We are upfront and clear about our contributions in the paper.
> > > > > > >
> > > > > > > Thank you again for your comments and suggestions, they have improved our paper.

---

### Author Rebuttal · Authors · 2023-08-10

Details about these experiments have been included in individual responses. Full details will also be included in the camera-ready version.

---

### Decision · Program_Chairs · 2023-09-21

**Decision:**

Accept (poster)

**Comment:**

This paper extends the PSRO algorithm to teams and provides some theoretical guarantees. The setting is such that members of a team cannot communicate with one another during the game. The reviewers' main concern was novelty, but the conversation in the rebuttal phase seems to have straightened that out.